# Interventions to address empathy-based stress in mental health workers: A scoping review and research agenda

Hannah May[1,2]*, Josie Millar[1], Emma Griffith[1,3], Chris Gillmore[3], Mhairi Kristoffersen[1], Ross Robinson[3], Michael West[4]

**1** Department of Psychology, University of Bath, Bath, United Kingdom, **2** St. George's University Hospitals NHS Foundation Trust, London, United Kingdom, **3** Avon and Wiltshire Mental Health Partnership MHS Trust, Bath, United Kingdom, **4** Management School, University of Lancaster, Lancaster, United Kingdom

* hannahrebeccamay@hotmail.co.uk

**Data Availability Statement:** The review protocol was registered on OSF and is available at https://osf.io/b7kcr/. The search terms used to conduct

## Abstract

Consistently engaging with client distress can negatively impact mental health workers (MHWs). This has been described by the concept of empathy-based stress (EBS) (which encompasses burnout; secondary traumatic stress; compassion fatigue and vicarious trauma). Previous reviews of interventions to reduce EBS have not addressed MHWs as a distinct group, despite evidence suggesting they are particularly vulnerable to it. In the context of rising demand for mental health services, it is especially important to understand how to mitigate the impact of EBS on MHWS. This scoping review therefore aimed to identify and describe available interventions to reduce or prevent EBS in MHWs. A systematic scoping review of the literature between 1970 and 2022 was undertaken using five electronic databases. A total of 51 studies were included, which varied significantly with regards to: interventions used; study methodology and theoretical underpinnings. Studies were grouped according to the level at which they aimed to intervene, namely: individual; team or organisational. The review concluded that most studies intervened at the level of the individual, despite the proposed causes of EBS being predominantly organisational. Furthermore, theoretical links to the origins of EBS were largely unclear. This suggests a lack of empirical evidence from which organisations employing MHWs can draw, to meaningfully prevent or reduce EBS in their staff. A dedicated research agenda is outlined to address this, and, other pertinent issues in the field and signifies a call for more theoretically grounded research.

## Introduction

### Empathy-based stress

People accessing mental health services have been found to have better outcomes if they experience compassionate [1] and empathic [2] relationships with staff caring for them. Clinicians themselves are also found to experience less distress and greater reward [3–5]. However,

the systematic search in each data base are included in supporting information. All papers identified in the systematic search were uploaded to Covidence for screening and extraction, available at https://app.covidence.org/reviews/130454. Characteristics of included studies (Table 4), studies reporting ethnicity data (Table 5) and descriptions of intervention categories from included studies (Table 6) are contained within the manuscript. The quantitative measures used in included studies (S3) and the recommendations for future research made in included studies (S4) are in supporting information. The Preferred Reporting Items for Systematic reviews and Meta-Analyses extension for Scoping Reviews (PRISMA-ScR) Checklist is included in supporting information (S1).

**Funding:** The author(s) received no specific funding for this work.

**Competing interests:** The authors have declared that no competing interests exist.

research has also identified potential costs to clinicians working with distressed clients. These include: burnout [6]; secondary traumatic stress (STS) [7, 8] compassion fatigue (CF) [7] and vicarious trauma (VT) [9]. Distinct definitions exist for each of the terms describing this problem (see Table 1). However, the extent to which they differ from one another is unclear and they are often used interchangeably in the literature [10, 11].

These inconsistencies led Rauvola et al [11] to propose an alternative conceptualisation in which VT, burnout, CF and STS are all highly related aspects of 'empathy-based stress' (EBS; Fig 1). EBS is described as applicable to anyone whose occupation exposes them to the distress of others and requires that they respond with engagement and empathy, and is therefore of particular relevance to MHWs. It is defined as "a stressor–strain-based process of trauma at work, wherein exposure to secondary or indirect trauma, combined with empathic experience, results in empathy-based strain and additional outcomes (i.e., other occupational health/strain outcomes; work affect, behaviours, and cognitions)" [11] (p299). In the EBS model; VT, STS and CF are the three varying forms of empathy-based strain that result from empathic engagement with the traumatic experiences of others. The state of burnout is conceptualised as a reaction to this strain.

EBS brings together established pre-existing concepts in a way that allows for distinct but inter-related terms to co-exist, thereby offering a potential solution to the conceptual ambiguities outlined above. Being dynamic, process-driven and applicable across professional fields, it is akin to established models in the occupational health literature, including the Job Demands-Resources model which posits that all jobs comprise interacting sets of psychologically taxing demands and rewarding job resources [17]. This review will henceforth use the term EBS, in reference to the dynamic model of "traumatic stressor exposure, empathic experience, and adverse reactions" (p. 297) [11].

**Table 1. Definitions of terms included in the empathy-based stress construct.**

| Term | Definition |
|---|---|
| **Burnout** Maslach [6] | A psychological response to chronic interpersonal stressors at work, resulting in exhaustion, detachment and feelings of incompetence. First identified in caregiving professions where services are provided via a relationship with people in need [12, 13]. These relationships (whilst rewarding) necessitate prolonged and intense emotional contact with clients which over time can produce burnout [13]. |
| **Secondary Traumatic Stress (STS)** Figley [7] Newell et al [14] | A form of second-hand post-traumatic stress disorder (PTSD). In STS, symptoms (e.g. hypervigilance, avoidance, intrusive thoughts/memories) arise out of exposure to the trauma of others, rather than through direct personal experience. |
| **Compassion Fatigue (CF)** Figley [7] | Figley [7] substituted STS with the term CF as a less stigmatising way of referring to the impact of these experiences on healthcare professionals. CF is "a state of exhaustion and dysfunction biologically, psychologically, and socially as a result of prolonged exposure to compassion stress" (p. 253). CF has an inverse relationship with CS; leading to the suggestion that overwhelming levels of CF may interfere with the capacity to experience CS [15]. |
| **Vicarious Trauma (VT)** Pearlman et al [9] | VT is defined as "the transformation that occurs in the inner experience of the therapist [or worker] [. . .] as a result of empathic engagement with clients' trauma material" [9]Whilst VT may result in PTSD-type responses (flashbacks, intrusions, dissociation [16], definitions emphasise the negative impact on the clinician's cognitive schemas [9]. |

*Note*: STS = secondary traumatic stress; CF = compassion fatigue; VT = vicarious trauma; PTSD = post-traumatic stress disorder

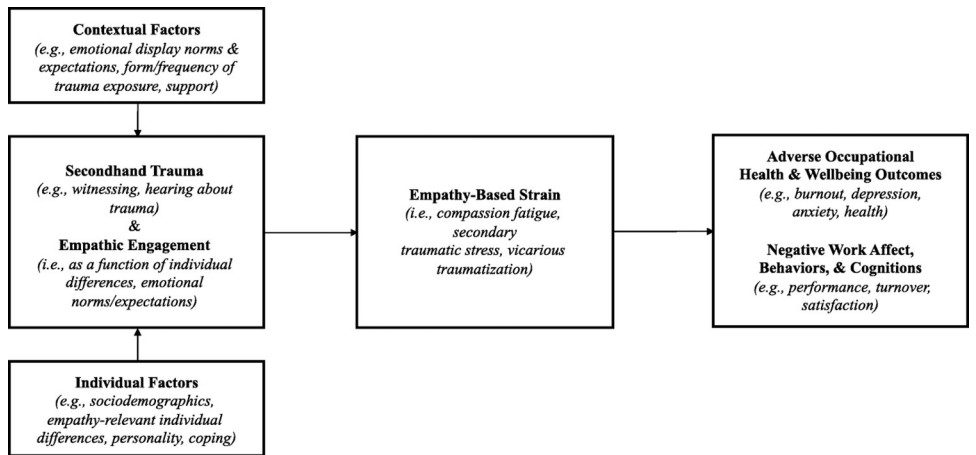

**Fig 1. Diagram of empathy based stress model from Rauvola et al [11], reproduced with permission).**

## Empathy based stress and mental health workers

There is evidence to suggest that EBS impacts care providers from diverse professional backgrounds [18]. However, Mental health workers (MHWs), are often required to engage explicitly with their clients' distressing experiences [19]. For example, successful psychotherapy is thought to depend in part on the therapist's ability to empathise with and absorb the pain clients are experiencing [20, 21]. This empathic engagement predicts better client outcomes [22]; however, leaves MHWs at increased risk of EBS [19, 23].

Rates of trauma exposure are higher amongst people accessing mental health services, with estimates suggesting that roughly half have experienced physical abuse, whilst over a third have experienced sexual abuse [24]. As services become increasingly aware of the importance of trauma-informed approaches to health and social care [25] MHWs are being encouraged to explicitly ask clients about potential traumatic experiences [26] Greater exposure to client trauma/distress is associated with higher CF in MHWs [10]which in turn is associated with a range of adverse outcomes including poor job satisfaction, absenteeism and poor patient care [27, 28]. This demonstrates that treatment and prevention of EBS is of importance not only to individual MHWs, but also to the organisations that employ them. NHS England [29] warn that increased demand for healthcare services leaves NHS staff vulnerable to compassion fatigue, which has been linked to depression, stress and chronic illness [19, 28, 30]. These are serious concerns for the workforce, as demonstrated in a recent NHS staff survey which found 44.8% of workers felt unwell due to work-related stress; a percentage that has been climbing steadily since 2016 [31].

## Current review and aims

Previous reviews targeting healthcare workers suggest that intervention/training may be effective in reducing EBS [32]. However, existing reviews have either amalgamated MHWs with other healthcare workers as one group -e.g., focusing purely on mindfulness and compassion interventions [33]—or excluded them [32]. Bercier et al [34] attempted to review EBS interventions for MHWs, however were unable to identify relevant studies and called for more research. Sutton et al [35] examined the impact of organisational factors on some specific forms of EBS. However, they did not consider other forms of intervention and only looked at particular professional groups.

This scoping review therefore aimed to review available interventions for preventing or treating EBS in MHWs. Given the urgency indicated by staff stress levels [31] and workforce shortages it seems timely to ask; "What interventions exist to prevent or treat EBS in MHWs?". Using systematic scoping methodology, we aimed to answer this question via the following four objectives:

1. To assess and summarise available interventions for preventing or treating EBS in MHWs.

2. To ascertain the theoretical underpinnings and assumptions of these interventions.

3. To assess how EBS has been measured.

4. To make recommendations for future research in terms of both treatment and prevention.

## Method

### Design

Scoping reviews offer a means to map out the scope and type of evidence available in a given area [36]. Scoping reviews can be conducted as self-contained reviews when there is insufficient knowledge to generate specific questions for a systematic review [37]. To this end, scoping reviews: have broader inclusion criteria compared to systematic reviews; do not assess bias in included studies; do not attempt to synthesise evidence or arrive at precise recommendations. Despite these differences, scoping reviews are *systematic* and include an *a priori* protocol and an exhaustive, systematic replicable search strategy [38, 39].

An a priori protocol was developed for the current scoping review and registered on Open Science Framework: https://osf.io/b7kcr/. The scoping review was conducted in accordance with the Joanna Briggs Scoping Review Framework [40] and is reported in line with the Prisma Extension for Scoping Reviews [41]. See S1 File for completed Preferred Reporting Items for Systematic reviews and Meta-Analyses extension for Scoping Reviews (PRISMA-ScR) Checklist.

### Electronic searches and search terms

Searches were conducted in the following databases: PubMed, PsychINFO (including APA PsycInfo, APA PsycArticles, APA PsycExtra and APA PsycTests), Embase, PTSDPubs. Grey literature was accessed via PsycExtra (theses) and the Cochrane Central Register of Controlled Trials (CENTRAL; unpublished trials). Studies available in English and conducted between 1970 (when the term burnout was introduced; [42] and January 2021 were included in the initial search. An updated search was subsequently carried out, incorporating papers published up to and including September 2022 (date range 2021 to 2022).

The search terms were determined by considering the core elements of the research question, in consultation with a research librarian (see Table 2). Given the inconsistent/overlapping definitions of terminology noted in the literature [10, 11], EBS search terms incorporated the empathy-based strain constructs [11] (VT, STS, CF) as well as burnout. Precise terminology, boolean operators and truncation were adapted/applied in accordance with each database (see S1 File for full search strategy). When full texts could not be retrieved via inter-library loan, effort was made to locate/contact authors. All identified papers were uploaded to Covidence: https://app.covidence.org/reviews/130454.

### Study selection

Title and abstract screening, as well as full-text review, was conducted by two independent reviewers (HM and MK/RR) according to the inclusion/exclusion criteria shown in Table 3.

**Table 2. Example of search terms.**

| Search item of interest | Provisional search terms |
|---|---|
| Empathy based stress | Burnout OR Compassion fatigue OR secondary traumatic stress OR STS OR vicarious trauma OR VT OR compassion stress |
| Mental health workers | mental health nurs* OR forensic nurs* OR counsellor* OR psychologist* OR psychiatrist* OR psychotherapist* OR therapist* OR mental health support worker* OR mental health social worker* OR support time and recovery worker* OR family therapist* OR therapist* OR CBT therapist* OR mental health occupational therapist* |
| Intervention | Treat* OR prevent* OR train* OR program* OR interven* |

Whilst the MHW search terms in Table 2 were used to guide the searches, some identified studies used different terms to describe participant job roles, or more general terms e.g., doctor, nurse. In these cases, the studies were carefully read in order to identify the nature of the client group, participants were working with, and/or the study setting. Such studies were included only if they clearly referenced client groups with mental health needs e.g., clients with psychosis, or healthcare settings that clearly had a mental health focus e.g., psychiatric wards.

## Charting the data

Information from included studies was charted by reviewers HM and MK, using the following headings:

**Characteristics of included studies.** *Study characteristics*. Authors, year of publication, study location, methodology, study design, study aims, EBS construct targeted, type of intervention/training, method of evaluating EBS.

*Sample characteristics*. Sample size, age, gender, setting, % of MHWs in sample, specific profession/role of MHWs.

**Characteristics of intervention/training.** This included: intervention/training type; the aims and rationales for the interventions/training programs administered; duration of intervention; and method of delivery. Information about the theoretical underpinnings of the intervention/training provided were recorded.

**Table 3. Criteria for study inclusion/exclusion.**

| | Inclusion Criteria | Exclusion Criteria |
|---|---|---|
| Population | At least 50% of participants are mental health workers (see Table 2 for included professions/synonyms). | Studies including other healthcare staff whose role does not have a mental health focus/the general public will be excluded. Children, adolescents |
| Concept | Studies involving any intervention (including organisational strategies e.g. increased supervision) or training program that aims to prevent or reduce EBS (including STS, CF, VT & burnout). Studies that in some way evaluate the impact of the intervention/training on EBS. | Studies that only assess MHW's individual differences and/or self-care strategies in relation to EBS. |
| Context | Study settings may include any setting where mental health workers are employed. | |
| Source | Primary research studies, systematic reviews and meta-analyses of any publication status. Qualitative or quantitative studies. | Theoretical papers, opinion pieces or letters to the editor Books, book chapters |

*Note.* EBS = empathy-based stress; MHWs = mental health workers; STS = secondary traumatic stress;

CF = compassion fatigue; VT = vicarious trauma.

**Outcomes.** Measures used to assess EBS and, where possible, outcomes on these measures were recorded. Any recommendations for practice or further research on training/interventions of this nature were documented.

## Results

A total of 51 studies were included in the scoping review. See Fig 2 for full details of the search outcomes. The results section initially reports on the characteristics of the included studies. Thereafter, for all included interventions the nature; length; theoretical underpinnings/ assumptions and means of measurement are synthesized narratively.

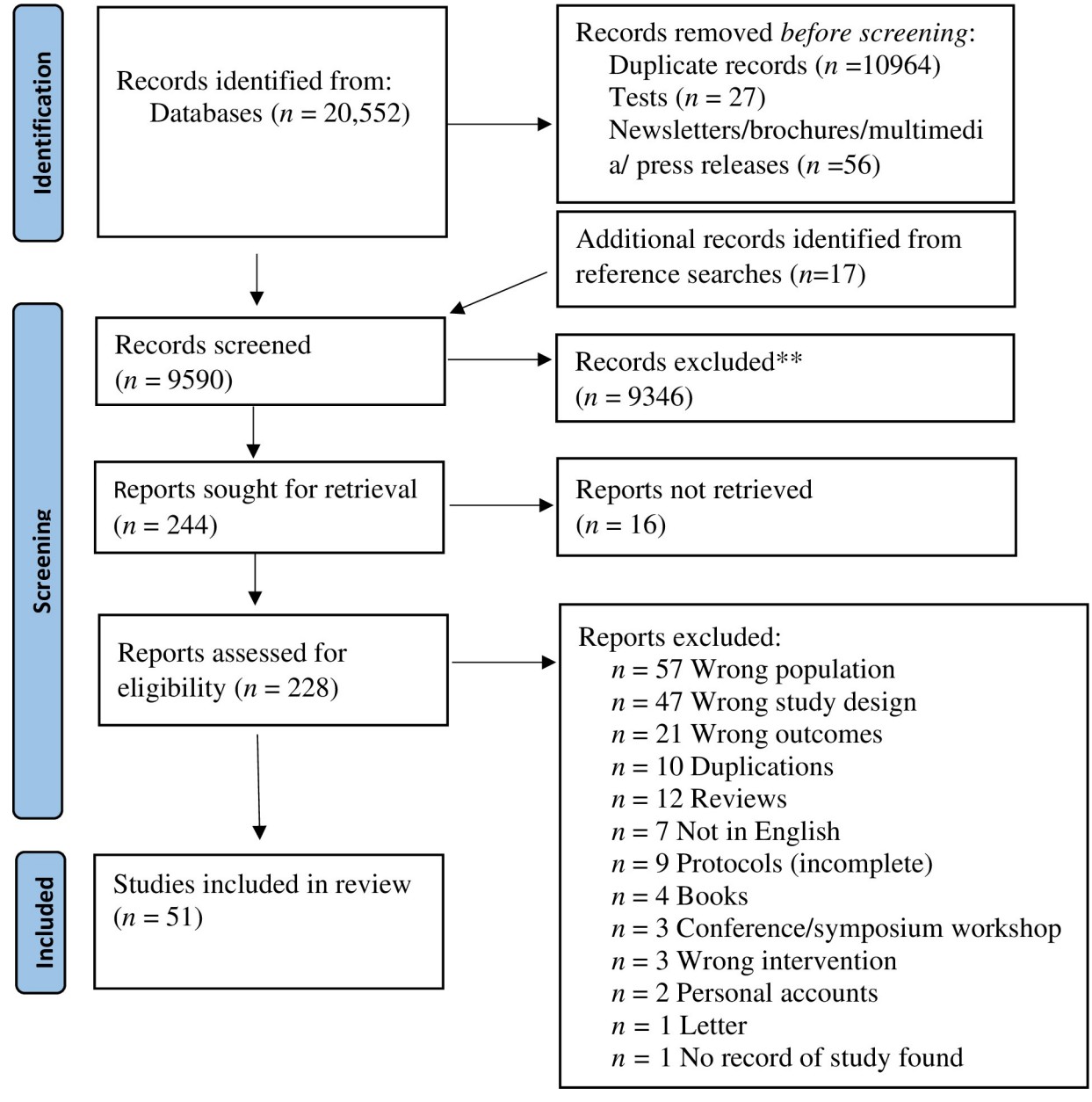

**Fig 2. PRISMA flow chart of the screening process.**

Finally, the relevant occupational health research and context is considering by the mapping of included interventions on to the three types of organizational intervention (i.e., primary, secondary and tertiary) for improving staff wellbeing and mental health described by Tetrick and Quick [43].

## Characteristics of included studies

Table 4 provides an overview of the key features of the 51 included studies. The year of publication ranged from 1981 to 2021. The majority of studies ($n = 28$) were conducted in the United States of America (USA). Most studies ($n = 43$) used a quantitative methodology. Study designs varied, ranging from case studies ($n = 1$) to randomised controlled trials (RCTs) ($n = 5$) with repeated measures designs ($n = 25$) being the most common. Samples varied in size from $N = 2$ to $N = 296$ and consisted of a diverse range of MHWs, most commonly nurses who comprised 100% of the sample in nine studies, and part of the sample in eight. Of the 40 studies that provided gender breakdowns: 27 comprised over 50% women; and seven were 100% women. Only 19 studies provided information on participant ethnicity (see S1 File), most of which had majority ($n = 14$) or entirely ($n = 1$) White samples. Other ethnicities represented included Black, Hispanic, Latino/Latina, Asian, Indian, Chinese, Malay, Mauritian, multi-racial, American Indian and Pacific Islander. Eight of the 20 studies reporting ethnicity described a proportion of participants as having 'other' or 'non-White' ethnicity, with no further information provided. A further four studies reported a percentage of White participants but provided no indication of the ethnic backgrounds of the rest of the sample. Settings included: community mental health ($n = 14$); inpatient ($n = 7$); outpatient/clinic ($n = 4$); multiple settings ($n = 3$); forensic ($n = 3$); mental health centres ($n = 2$); substance use services ($n = 2$); veteran's services ($n = 1$); primary school ($n = 1$); domestic abuse shelter ($n = 1$); child advocacy centre ($n = 1$). Most studies targeted a single aspect of EBS ($n = 47$), whilst some targeted multiple ($n = 4$). Burnout was by far the most common outcome targeted ($n = 41$) although STS ($n = 7$) and CF ($n = 8$) were also addressed. According to the EBS model [11] this means that over 80% of identified studies were targeting one of the adverse outcomes of empathy-based strain, rather than addressing empathy-based strain itself. Most studies ($n = 46$) were classified as treatment studies, in that, as far as could be determined, they aimed to examine the ability of an intervention to treat/impact upon EBS, with the assumption apparently being that EBS was already present. Five studies aimed to both treat and prevent EBS. No studies targeted prevention alone.

## Summary of available interventions for preventing or treating EBS in MHWs

Studies were included if they evaluated an intervention for reducing/preventing EBS that was externally facilitated (i.e., the organisation/researchers provided and arranged it). Studies measuring internally facilitated or 'self-care' behaviours (or other individual attributes) were excluded on the basis that it is primarily an organisational responsibility, not an individual one, to tackle EBS. Two included studies challenged these criteria. The employee wellness program [58] was organised and coordinated by the researchers, however relied on employees to deliver the wellness interventions and took place during employee's lunch hour. The guided imagery study [67] provided participants with guided imagery tracks on MP3 players and instructed them to listen to them three times a week over the intervention period, during work breaks. These studies were included because of the external facilitation by researchers in guiding participants on when and how to perform the interventions. Similar studies that provided

**Table 4. Characteristics of included studies.**

| ID | Study | Country | Publication Status | Study Design | EBS Construct Targeted | Setting | N | % of MHWs (specific role) | Gender | Intervention |
|---|---|---|---|---|---|---|---|---|---|---|
| 01 | Haynos et al [44] | USA | PRJ | Quantitative Repeated measures | Burnout | CYP inpatient | 22 | 100% (MH nurses) | Not Given | DBT skills coaching training |
| 02 | Roberts [UP] | UK | UT/D | Quantitative Repeated measures | Burnout | Not given | 141 (39 at FU) | Not given | 79.4% women | DBT skills training |
| 03 | Clarke et al [45] | UK | PRJ | Quantitative RCT | Burnout | Inpatient, outpatient | 140 (57 at FU 1; 36 FU 2). | 100% (Staff working with clients with BPD) | Not given | ACT Training vs psycho-ed. BPD training[1] |
| 04 | Doyle et al [46] | UK | PRJ | Quantitative Experimental design | Burnout | Forensic | 26 | 61.50% (Forensic nurses) | 73.1% women | Psychosocial intervention training |
| 05 | Hallberg [47] | Sweden | PRJ | Mixed methods Repeated measures | Burnout | CYP inpatient | 11 | 100% (Child MH nurses) | 63.3% women | Systematic clinical supervision |
| 06 | Hunnicutt & MacMillan [48] | USA | PRJ | Quantitative Experimental design (2 intervention groups; 1 control) | Burnout | Community | 251 | Not given | Not given | Burnout workshop standalone vs with staff consultation. |
| 07 | Landis, [UP] | USA | UT/D | Mixed methods Repeated measures | ST | Community | 5 | 100% (SWs) | 80% women | CF/ST workshop; 'Sharevision' meetings |
| 08 | Perseius et al [49] | Sweden | PRJ | Mixed methods Repeated measures | Burnout | Adult & CYP psychiatry clinics | 22 | 100% (Drs; psychol; MH nurses/care assistants; therapist, OT) | 86.4% women | DBT training |
| 09 | Raab et al [50] | Canada | PRJ | Quantitative Repeated measures | Burnout | Community | 22 | 100% (MH care workers) | 100% women | MBSR |
| 10 | Razzaque et al [51] | UK | PRJ | Mixed methods Repeated measures | Burnout | Community | 26 | 100% (Psychr) | 65.4% women | Mindfulness Retreat |
| 11 | Rollins et al [52] | USA | PRJ | Quantitative Experimental design | Burnout | Inpatient & outpatient, (mostly veterans services) | 145 | 100% (Behavioural health providers) | 71% women | 'BREATHE' burnout reduction workshop |
| 12 | Melchior et al [53] | The Netherlands | PRJ | Quantitative Experimental design | Burnout | Inpatient | 161 | 100% (MH nurses) | 72% women | Primary nursing |
| 13 | Duckworth [54] | USA | UT/D | Quantitative Repeated measures | STS | CYP community | 16 | 100% (psychol; nurses; SWs, therapists; admin) | 93.8% women | STS education program and prevention plan |
| 14 | Boone [UP] | USA | UT/D | Quantitative Experimental design | STS | Not given | 97 (53 at FU) | 100% (Domestic violence counsellors) | 88.7% women | Poetry writing therapy |
| 15 | Flarity et al [55] | USA | PRJ | Quantitative Repeated measures | CF | Inpatient (emergency department) | 7 | 100% (Forensic nurses) | 100% women | CF prevention and resilience training |

(*Continued*)

**Table 4.** (*Continued*)

| ID | Study | Country | Publication Status | Study Design | EBS Construct Targeted | Setting | N | % of MHWs (specific role) | Gender | Intervention |
|---|---|---|---|---|---|---|---|---|---|---|
| 16 | Suyi et al [56] | Singapore | PRJ | Quantitative Repeated measures | Burnout | Not given | 37 | 100% (Psychol; Psychr; SWs) | 81.1% women | MBSR |
| 17 | Kovač et al [57] | Slovenia | PRJ | Quantitative Experimental design | Burnout | School | 30 | 100% (School counsellors) | 100% women | Supervision (relational family model) |
| 18 | Van Kirk [58] | USA | PRJ | Quantitative Repeated measures | Burnout, STS & CS | Veteran MH services | 57 | 100% (nurses, SWs, psychol, psychr, counsellors, SA therapists; OTs; assistants) | 64% women | Employee wellness program |
| 19 | Alenezi et al [59] | Saudi Arabia | PRJ | Quantitative Experimental design | Burnout | Inpatient | 296 | 100% (MH nurses) | 49.3% women | Burnout prevention programme |
| 20 | Eriksson et al [60] | Sweden | PRJ | Quantitative RCT | Burnout | Not given | 101 (81 at FU) | 100% (Psychol) | 96% women (plus one NB person) | Web-based compassion program |
| 21 | Wymer [UP] | USA | UT/D | Quantitative Single case research design | STS | Child advocacy centres | 3 | 100% (CSA counsellors) | 100% women | Affective check-in supervision |
| 22 | Chilton et al [61] | USA | PRJ | Qualitative Case study | CF & STS | Addiction treatment centre. | 2 | 100% (art/expressive arts therapists) | 100% women | El Duende 'Process Painting' |
| 23 | Askey-Jones [62] | UK | PRJ | Quantitative Repeated measures | Burnout | Primary & secondary | 69 (43 at FU) | 52% (MH Nurses) | 76% women | MBCT group therapy |
| 24 | Ewers et al [63] | UK | PRJ | Quantitative Experimental design | Burnout | secure forensic settings | 33 (20 at FU) | 100% (MH nurses) | 51.5% women | Psychosocial Intervention Training |
| 25 | Finamore et al [64] | UK | PRJ | Quantitative Repeated measures | Burnout | Not given | 253 (201 at FU) | 100% Mixed | Not given | Knowledge & Understanding Framework PD Training |
| 26 | Reyes Ortega et al [65] | Mexico | PRJ | Quantitative Experimental design | Burnout | BPD Clinic | 6 | 100% (Psychol; psychr) | 50% women | 'Helping the Helper' social connectedness intervention |
| 27 | Brady et al [66] | USA | PRJ | Quantitative Repeated measures | Burnout | Inpatient | 16 | 100% (Psychr; Psychol; MH technicians Activity Therapists) | Not given | MBSR |
| 28 | Walker, [UP] | USA | UT/D | Quantitative Repeated measures | Burnout | Can't access-preview | 34 | 100% (Can't access-preview) | Can't access-preview | MBSR |
| 29 | Kiley et al [67] | USA | PRJ | Quantitative RCT | CF | Community | 69 (56 at FU) | 100 (SWs; counsellors; psychr; CMs, support staff and management) | Not given | Guided Imagery |

(*Continued*)

**Table 4.** (Continued)

| ID | Study | Country | Publication Status | Study Design | EBS Construct Targeted | Setting | N | % of MHWs (specific role) | Gender | Intervention |
|---|---|---|---|---|---|---|---|---|---|---|
| 30 | Riley et al [68] | USA | PRJ | Quantitative 2 experimental conditions, no control | STS, CF & Burnout | Multisite MH centre | 38 (28 at FU 1; 25 at FU 2; 19 at 6 month FU) | 100% (Not given) | 84.2% women | Yoga-based stress management vs cognitive-behavioural stress management |
| 31 | Rosada et al [69] | USA | PRJ | Quantitative Repeated measures cross-over design | Burnout | Community | 45 | 100% (MH clinicians) | 73.3% women | Reiki (reiki vs 'sham') |
| 32 | Ifrach & Miller [70] | USA | PRJ | Quantitative Repeated measures | CF | Domestic abuse shelters | 30 | 100% (DV counsellors) | 100% women | Social action art therapy |
| 33 | Carmel et al [71] | USA | PRJ | Quantitative Repeated measures | Burnout | Community | 9 | 100% (MH practitioners; SA counsellors) | 88% women | DBT training |
| 34 | Newman et al [8] | South Africa | PRJ | Qualitative Retrospective | Burnout | CYP Community | 30 | 100% (Child Psychr, psychr, Drs, nurses, clinical psychol, OTS, admins, interpreter) | 70.6% women | Drumming |
| 35 | Paulson et al [72] | USA | PRJ | Mixed methods Repeated measures | Burnout | Community (rural) | 6 | 100% (psychol; counsellors) | Not given | Online peer consultation network (support group) |
| 36 | Luoma & Vilardaga [73] | USA | PRJ | Quantitative Experimental design | Burnout | Community | 20 | 80 (Psychol; Counsellors) | 40% women | ACT Training |
| 37 | Little [74] | USA | UT/D | Quantitative Repeated measures | Burnout | Not given | 37 | 100% (Therapists; CMs; admins) | 75.7% women | DBT training |
| 38 | Scarnera et al [75] | Italy | PRJ | Quantitative Repeated measures pilot study | Burnout | Not given | 25 | 100% (Nurses, educators; psychol; psychr; managers) | 44% women | Interpersonal relationship management training |
| 39 | Ray [UP] | USA | UT/D | Quantitative Experimental design | Burnout | Community | 36 | 100% (SWs; psychol; therapists; attendant; nurses; admin) | 75% women | Stress management training program |
| 40 | Jones [76] | UK | PRJ | Quantitative Repeated measures | Burnout | Inpatient | 72 (62 at 3rd FU, 49 at 4th FU) | 100% (MH nurses; HCAs; OTs ward managers; CNSs) | 57% women | Interventions for psychosis training + clinical practice development |
| 41 | Redhead et al [77] | UK | PRJ | Quantitative RCT | Burnout | Forensic inpatient | 42 | 100 (Forensic nurses) | Not specified | Psychosocial intervention training |
| 42 | Salyers et al [78] | USA | PRJ | Quantitative Experimental design | Burnout | Substance abuse | 84 (74 at FU) | Not given | 87% women | BREATHE Burnout reduction retreat |
| 43 | Weingardt et al [79] | USA | PRJ | Quantitative RCT (2 experimental conditions, no control) | Burnout | Outpatient & controlled settings | 147 | 100% (SA counsellors) | 62.1% women | Online CBT training + supervision meetings (high vs low fidelity) |

(*Continued*)

**Table 4.** (Continued)

| ID | Study | Country | Publication Status | Study Design | EBS Construct Targeted | Setting | N | % of MHWs (specific role) | Gender | Intervention |
|---|---|---|---|---|---|---|---|---|---|---|
| 44 | Leykin et al [80] | USA | PRJ | Quantitative 2 experimental conditions, no control | Burnout | Community | 149 (112 at 1st FU 81 at 2nd FU) | 100% (SA counsellors) | 62.6% women | Online training in CBT for substance abuse + supervision meetings (high vs low fidelity) |
| 45 | Gentry et al [81] | USA | PRJ | Quantitative Repeated measures | Burnout & CF | Not given | 83 | 100% (SWs; counsellors; psychols) | Not given | Certified CF Specialist Training |
| 46 | Hayes et al [82] | USA | PRJ | Quantitative Experimental design (2 experimental conditions; 1 control) | Burnout | Not given | 93 | 100 (SA counsellors) | 63% women | ACT training vs Multicultural training |
| 47 | Anderson [83] | USA | UT/D | Quantitative Experimental design | Burnout | MH centre | 40 | 100% (counsellor-therapists) | 60% women | Facilitator-led peer groups |
| 48 | Mehr et al [84] | USA | PRJ | Quantitative Repeated measures | Burnout | Community | 27 | 100% ('MH workers') | 100% women | Stress-reduction/ positive imagery conference + follow up meetings |
| 49 | Ballew, [UP] | USA | UT/D | Quantitative Quasi-experimental control time series design | CF | Not specified | 43 (19 at FU) | 68% ('MH professionals') | 76.9% women (2 people identified with multiple genders) | Professional Resilience and Optimization workshop |
| 50 | Chochol et al [85] | USA | PRJ | Mixed methods Repeated measures | Burnout | Child and adolescent psychiatry | 6 (pilot study) | 100% (Child and adolescent psychiatry fellows) | Not specified | Balint-like group incorporating brief emotional awareness modules |
| 51 | Bartels-Velthuis et al [86] | Netherlands | PRJ | Quantitative Experimental design | CF | Outpatient MH clinic | 47 | 100% (doctors; psychols; nurses; mindfulness teacher; SW; MH counsellor; physiotherapist; drama therapist) | 91.5% women | Interpersonal Mindfulness Program + 45–60 min daily home practice |

*Note*: ACT = acceptance and commitment therapy; BPD = borderline personality disorder; CF = compassion fatigue; CM = case managers; CNS = clinical nurse specialist; CS = compassion satisfaction; CSA = child sexual abuse; CYP = children and young people; DBT = dialectical behaviour therapy; FU = follow up; HCA = health care assistants; PD = personality disorder; PRJ = peer reviewed journal; MBCT = mindfulness-based cognitive therapy; MBSR = mindfulness-based stress reduction; MH = mental health; MSc = masters NB = non-binary; OT = occupational therapist; PD = personality disorder; psychr = psychiatrist; psychol = psychologist; RCT = randomised controlled trial; SA = substance abuse; ST = secondary trauma; STS = secondary traumatic stress; SW = social worker; UM/T = unpublished masters/thesis; UP = unpublished

[1] The primary aim of this RCT was to compare the effects of these two interventions on stigma towards clients with BPD; however clinician burnout was also measured

a potential moderator of EBS without this structure were excluded, e.g., a case study (*N* = 1) gave the participant a mindfulness app but no guidance on when and how to use this [87].

Given the diverse range of interventions described, for clarity, they have been grouped according to the main intervention components as described in the studies themselves (Table 5). It is

**Table 5. Descriptions of the broad categories of interventions used in identified studies.**

| Intervention | Description |
|---|---|
| **Training in a therapeutic modality (*N* = 11)** | • Dialectical Behaviour Therapy (DBT; *n* = 5)<br>• Psychosocial Intervention Training (PIT; *n* = 3)<br>• Cognitive Behavioural Therapy (CBT; *n* = 2)<br>• Specific training on understanding/ working with psychosis (*n* = 1)<br>• Knowledge & understanding framework (KUF) for working with borderline personality disorder (BPD; *n* = 1; compared with ACT—see below). |
| **EBS training and prevention/ Resiliency training (*N* = 7)** | • Education on causes, signs and impacts of EBS & encouragement to develop coping/prevention strategies.<br>• Opportunities for participants to practice some of these strategies e.g. breathing exercises; progressive muscle relaxation; developing secondary trauma narratives (*n* = 3)<br>• Resiliency training, incorporating EBS education and development of coping strategies, as well as teaching resiliency skills (e.g. self-regulation, self-care) to fight off compassion fatigue (*n* = 1)<br>• Training MHWs to deliver the certified compassion fatigue specialist training (CCFST [81]) to other MHWs experiencing CF hypothesizing there would be a 'training-as-treatment' impact on participants' own CF (*n* = 1) |
| **Mindfulness (*N* = 7)** | • Mindfulness-based stress reduction (MBSR; *n* = 4)<br>• Mindfulness-based cognitive therapy (MBCT; *n* = 1)<br>• Mindfulness-based professional development (MBPD; *n* = 1)<br>• Retreat taught participants mindfulness theory and practice and encouraged workers to develop their own practices (*n* = 1) |
| **Wellness/stress reduction (*N* = 6)** | • Training in stress management techniques (i.e., identifying stressors & practicing stress-reduction skills, including guided imagery and progressive muscle relaxation (*n* = 2)<br>• Guided imagery (*n* = 1)<br>• Reiki (*n* = 1)<br>• wellness program(including mindfulness, nutrition and movement) (*n* = 1)<br>• yoga-based stress management (YBSM; combining yoga and aspects of cognitive stress management) Vs cognitive-behavioural stress management (*n* = 1) |
| **Peer support/ relationships (*N* = 4)** | • Facilitation of regular peer groups (*n* = 2; one focused on sharing thoughts and emotions regarding burnout issues, the other on case consultation)<br>• Fostering emotional closeness between MHWs using a functional psychotherapeutic approach (*n* = 1)<br>• Teaching participants techniques for handling professional interpersonal relationships (*n* = 1) |
| **Expressive arts (*N* = 4)** | • Art therapy (*n* = 1)<br>• Art therapy supervision (*n* = 1; included here, rather than in supervision, below, because creating art was the central component of the intervention)<br>• Poetry therapy (*n* = 1)<br>• Drumming (*n* = 1) |
| **Supervision (*N* = 3)** | • Psychodynamic group supervision (*n* = 1)<br>• relational family supervision (*n* = 1)<br>• affective check in supervision (*n* = 1)<br>• *It should be noted that whilst two studies introduced supervision to MHWs who had not previously been receiving it, the affective check in study recruited participants already receiving regular supervision and trained their supervisors to use the affective check in technique.* |
| **Acceptance and Commitment Therapy (ACT; *N* = 3)** | • Teaching MHWs ACT skills for personal use (*n* = 3; two studies comparing ACT with other interventions including BPD training (see above) and multicultural training) |

(*Continued*)

**Table 5.** (Continued)

| Intervention | Description |
|---|---|
| **BREATHE (*N* = 2)** | • Delivery of The BREATHE (Burnout Reduction: Enhanced Awareness, Tools, Hand-outs, and Education) program [78], incorporating burnout prevention information with practices including mindfulness, social (e.g. support structures and boundaries), physical (e.g. body scan), cognitive (e.g. thought challenging; identifying values), imagery, and other self-care activities. |
| **Primary nursing (*N* = 1)** | • Nursing intervention whereby both psychiatric and practical nurses were assigned to patients as primary nurse caregivers. |
| **Balint-like group (*N* = 1)** | • Sharing of a challenging case by one group member, after which group attendees asked clarifying questions. For approximately 10 min, the other group members and the group facilitators process the case and their emotional responses. Afterwards, the presenter returns to the group and has the opportunity to respond to the group's commentary, stimulating further discussion until the end of the hour. |
| **Compassion enhancement program (*N* = 1)** | • Standard mindfulness exercises such as breathing anchor and body scans, and compassion-focused exercises such as loving-kindness, and exercises of compassion with self and others. |

*Note*: ACT = acceptance and commitment therapy; BPD = borderline personality disorder; MHWs = mental health workers.

acknowledged that some of the more integrative interventions contained various elements and could potentially be categorised multiple ways.

The length of the interventions varied considerably. The majority (*n* = 36) were delivered over weeks/months, with timeframes ranging from two weeks to 12 months. Some studies were delivered in workshops/retreats of less than one day (*n* = 3), one (*n* = 3), two (*n* = 6) and three (*n* = 1) days in duration, with one intervention described as two to three days long. For studies where the total hours of intervention received could be calculated, this ranged from two to 104 hours. Six studies required 'homework' activities in-between or in addition to attending the interventions, including meditation practice and unfacilitated peer groups.

## Theoretical underpinnings and assumptions

The theoretical rationale for the interventions used to address EBS in MHWs, can be considered in two ways. Firstly, the route via which studies attempted to intervene and what this reveals about their underlying assumptions regarding the cause and nature of EBS. Secondly, there are the theoretical frameworks/rationales underpinning some of the specific interventions used.

**Route of intervention.** We have described the routes of intervention taken by the included studies as: 1. individual (personal practices; awareness-raising and role training); 2. team and 3. organisational.

*Individual*. Individual interventions intervened at the level of the individual to help them manage the impact of EBS and/or other demands associated with work. There were three subtypes of individual intervention: personal resilience; awareness-raising and role-training. Subsections of the individual route included:

• Personal resilienceinterventions: whichsought to provide MHWs with treatment and/or train them in using personal practices to manage their own negative experiences. These included: wellness/stress reduction; mindfulness; ACT; compassion enhancement; expressive arts and aspects of the BREATHE intervention and resiliency workshops. Awareness-raising interventions: which attempted to alleviate EBS by making MHWs aware of and

|  |  | Primary | Secondary | Tertiary |
|---|---|---|---|---|
| **Individual** | Personal Practice |  | • Wellness/stress reduction<br>• Mindfulness<br>• ACT<br>• Compassion enhancement<br>• Expressive arts<br>• Aspects of the BREATHE intervention and resiliency workshops |  |
|  | Awareness raising |  | • EBS prevention<br>• Aspects of the BREATHE intervention and resiliency workshops |  |
|  | Role training |  | • Training in a therapeutic modality (DBT; PSI; CBT) |  |
| **Team** |  |  | • Supervision<br>• Peer groups/relationships<br>• Balint-like group |  |
| **Organisational** |  | • Primary nursing |  |  |

**Fig 3. Intersection of level of intervention with type of organisational intervention.**

prepared to tackle the negative effects of EBS. These included: EBS prevention and aspects of the BREATHE intervention and resiliency workshops. Role-training interventions: which provided MHWs with training in therapeutic modalities or approaches to use with their clients. These included: training in a therapeutic modality.

*Team*. Team interventions were those focused on team relationships and supportiveness. These included: supervision; peer groups/relationships and Balint-like groups.

*Organisational*. Organisational interventions are those in which changes to the structure/running of the organisations employing MHWs were made. These included: primary nursing.

Fig 3 displays how these levels of intervention intersect with the three types of organisational intervention for improving staff wellbeing and mental health [43]: i) primary interventions aim to decrease or remove stressors at the organisational level i.e., at the source. They require changes to workplace practice and usually involve employees in the processing of developing interventions; ii) secondary interventions attempt to alter the individual's perception of, or responses to a stressor. They enable work-related stressors to be swiftly identified and attempt to mitigate these by increasing employees' coping skills, awareness and knowledge e.g., through additional training; iii) tertiary interventions are reactive and focus on

rehabilitation of those already experiencing significant strain resulting from stressors e.g., psychological therapy or occupational health services [43]. This demonstrates that all but one of the interventions were of the secondary type although they were operating at various levels of individual and team intervention.

**Specific theories and rationale.** Reference to theory was inconsistent, with some studies referring to an evidence base rather than theoretical concepts. Where theoretical underpinnings for interventions were made explicit, these are highlighted below (grouped by type of intervention). Evidence base/rational given for interventions not explicitly linked to theory are summarised in Table 6.

*Psychosocial Intervention Training (PIT)*. Clients with serious mental health difficulties may be perceived as troublesome, hard to understand and difficult to help and these perceptions may make MHWs feel demotivated [88] and burnt out [89]. PIT gives MHWs skills to empathically understand and intervene more effectively with these clients, which may increase their sense of reward and efficacy in their role and decrease burnout.

*EBS education & prevention/resiliency training*. One study [54] highlighted that secondary traumatic stress disorder is thought to occur when second-hand trauma is not integrated. Thus, teaching MHWs about this may enable them to better respond to their experiences. Salyer et al [78] combined burnout-reduction principles such as boundary setting with mindfulness techniques. For the study teaching MHWs to deliver Certified Compassion Fatigue Specialist Training (CCFST); it was hypothesised that this may reduce their own CF due to exposure to theories of EBS they learnt in the process [81].The study utilising resiliency training delivered a workshop titledthe Professional Resilience and Optimization workshop [90]—which is based in the same recovery program as the CCFST, the Accelerated Recovery Program [91]. The Accelerated Recovery Program is a manualised treatment for people experiencing compassion fatigue. The proposed effectiveness of the Professional Resilience and Optimization workshop was linked to various separate mechanisms that did not constitute a unified theory and are summarised in Table 6. However, the study utilising this program made reference to emotional contagion theory [7, 21] and proposed that the resiliency skills being taught would act as 'antibodies' of CF that would allow participants to resist its effects Ballew [Unpublished].

*Mindfulness/compassion*. Mindfulness and self-compassion encourage non-judgmental attitudes towards experiences and may therefore decrease unhelpful coping strategies and increase willingness to accept and experience negative emotions that might arise [92].

*ACT*. Key components of ACT (e.g., psychological flexibility, cognitive de-fusion and acceptance) may reduce impact and believability of negative thoughts/ feelings arising from difficulties/stress in working with clients [93, 94].

*Supervision*. Supervision is theorized to decrease, manage or prevent burnout as support systems increase MHWs ability to cope with work stress [95, 96]. Secondary trauma responses in trauma counsellors could be mitigated by supervision practices that validate their experiences of being personally impacted by their work as well as giving them skills to manage this [19, 97, 98].

*Arts therapy*. Expressive arts therapies can alleviate compassion fatigue [9] and their inclusion in MHWs' supervision can increase self-awareness and reduce stress e.g. [99, 100]. The study using drumming highlighted that this has been used in cultural healing practices since the beginning of ancient history with positive physiological and psychological effects [101]. The study using poetry referenced how writing may help in the integration of traumatic memories [102].

*Wellness/stress reduction*. This group of studies provided a diverse range of rationales for their interventions, which largely were not linked to explicit theories and are captured in

**Table 6. Evidence base provided by studies without an explicit theoretical basis for intervention.**

| Intervention Type | Evidence Base | |
|---|---|---|
| CBT (plus supervision) | The two studies training MHWs in CBT skills emphasised the format of training delivery and supervision above the content, i.e. highly structured, inflexible methods vs flexible and responsive training and supervision. They drew on research findings that indicate that supportive organizations that promote flexibility and autonomy may reduce job burnout [104, 105]; whereas highly centralised management practices are associated with higher levels of emotional exhaustion among counselling staff [106]. | |
| Primary nursing | No research explicitly investigated the impact of primary nursing (PN) on burnout. However, it was hypothesised that it might alleviate it given that low autonomy is associated with higher burnout in nurses [107], and PN increases autonomy [108] | |
| Peer/relationships | Studies noted that social support is negatively correlated with burnout [109] and that sharing one's emotions with others who are also experiencing difficulties can be therapeutic [110–112]. In addition, lack of reciprocity amongst colleagues and unhelpful comparisons with peers seem to have a part in the development of burnout [30]The study utilising a social connectedness intervention based this on the Interpersonal Process Model (IPM; [113], wherein a turn-by-turn relational process is thought to establish psychological intimacy. They highlight evidence suggesting that when members of a dyad engage in reciprocal self-disclosure and respond to one another with care and validation, this creates feelings of closeness and intimacy between them [114, 115]. | |
| DBT | Within the DBT model, the behaviours of clients with BPD that increase risk of burnout in MHWs (e.g. excessive contact with MHWs, demands for more regular therapy or threats towards MHWs) are considered therapy-interfering behaviours [116]. The model therefore incorporates a consultation group which aims to support and encourage MHWs, whilst maintaining their adherence to the model and identifying when clients are pushing boundaries so that this can be swiftly addressed. This is thought to reduce risk of burnout. In addition, emphasis on DBT clinicians practicing the core DBT skills of distress tolerance, mindfulness, emotion regulation, and interpersonal effectiveness is thought to be protective against burnout [116, 117]. | |
| Balint-like groups | The study utilising Balint-like groups cited research demonstrating that these groups target various outcomes including burnout and wellbeing [118]. In addition, it was noted that physicians have reported the groups decrease feelings of isolation and support their processing of emotional interactions [119] as well as increasing perceptions of social support for physicians working in palliative care [120]. | |
| Wellness/stress reduction | Stress Reduction workshop + follow up meetings | The study employing stress-reduction/positive imagery workshops highlighted that stress-reduction workshops can alter the course of occupational burnout via integrative methods including practical guidance and emotional support [95]. However, they point out that workshops alone are insufficient and frequent follow up is needed to maintain the changes in coping styles that can be introduced [121] |
| | Wellness programme | This study suggested that the energy resulting from compassion fatigue can positively impact compassion satisfaction if handled differently [122]. They cite findings that reduction in CF and gains in compassion satisfaction have been demonstrated for: Expressive writing [123], guided imagery [67], yoga and mindfulness [124], and music therapy [125]. |
| | Yoga-Based Stress Management | The study authors had devised this novel method through combining yoga with aspects of cognitive stress management. They cite research that yoga and cognitive stress management programs are associated with improved in wellbeing and stress reduction [126]. The authors also suggested the intervention was similar to MBSR and created opportunities to practice mindfulness (unreferenced in paper). |
| | Guided Imagery | The study using guided imagery cited findings that imagining an activity produces similar physiological reactions to actually carrying it out [127], and therefore suggested that calming and peaceful imagery may produce similar responses to real situations of tranquility. |

(*Continued*)

**Table 6.** (Continued)

| Intervention Type | Evidence Base | |
|---|---|---|
| | Reiki | The study using Reiki described how this energy-based treatment promotes energetic balance in recipients that holistically addresses emotional, physical and spiritual aspects of their self to balance and heal their energy [128] |

*Note.* CBT = cognitive behavioural therapy; CF = compassion fatigue BPD = borderline personality disorder; DBT = dialectical behaviour therapy; MBSR = mindfulness based stress reduction; MHWs = mental health workers; PN = primary nursing

Table 6. However, one stress management study highlighted work by [103] suggesting that people can be trained to observe how environmental stressors impact their behaviour and cognitions and thus choose more adaptive responses to stress.

**Measures.** An overview of the quantitative measures of EBS used in included studies is provided in S1 File. Variations of the Maslach Burnout Inventory were most commonly used (*n* = 28] consistent with the fact that burnout was the most common outcome targeted. Of the studies utilising qualitative/ mixed methods (*n* = 8), qualitative data collection was done via interviews (*n* = 4); free text/semi-structured questionnaires (*n* = 3); art created by participants (*n* = 1); focus group (*n* = 1) and narrative feedback (*n* = 1).

The majority of studies (*n* = 49) collected EBS measures both pre and post intervention. Most of these (*n* = 30) only collected follow up data at one time point. Of these 30 studies, 25 collected post measures within two weeks of the intervention end. However, some (*n* = 6) collected post measures between five weeks and 12 months post intervention. Four studies comprising long-term interventions (supervision; DBT training) took multiple measures throughout intervention phases. Twelve studies collected additional follow up data after the initial post data, either once (*n* = 11) twice (*n* = 2) or three times (*n* = 1). Timeframes for additional post-intervention data ranged from two to 18 months. Finally, one study [52] collected their first post-intervention data at six weeks following intervention, and then a further follow up at six months.

## Discussion

This scoping review is, to our knowledge, the first to describe the available evidence on interventions for EBS in MHWs as a distinct professional group. In addition, it considered the theoretical underpinnings for available interventions; determined how EBS has been measured; and developed a research agenda for future research. The findings relating to each of these aims are discussed below, with reference to the wider literature. Overall, the group of studies included in this review were heterogeneous and differed substantially with regards to: the interventions used; the length and delivery of interventions and the rationale for applying them. We discuss how the lack of consistency across these areas, coupled with the conceptual difficulties in the field, create issues for EBS intervention literature. In response, we put forth a research agenda with recommendations for how future researchers might navigate and address these difficulties.

### Available interventions for EBS

The overwhelming focus on burnout (*n* = 41) in included studies is notable. According to the EBS model proposed by Rauvola et al [11] STS, CF and VT are all forms of empathy-based strain that drive the process of EBS, while burnout is one of the resulting adverse outcomes.

The majority of interventions described in this review are therefore targeting an end result of EBS, rather than addressing the active constructs. Similarly, all studies aimed to address EBS already present in MHWs (rather than aiming to prevent it). This suggests a recognition of the scale of the problem in the mental health work force [29, 31], and suggests an emphasis on treatment over prevention.

## Participants

There was considerable variation in the specific roles/professions of MHWs, with nurses being the profession most represented. Other professions included psychiatrists, psychologists, social workers, expressive arts therapists, occupational therapists and health care assistants. Thus, the collective sample of participants differed considerably with regards to the nature and amount of training MHWs would have received to undertake their role.

Of further importance was participant ethnicity, which was reported by only 39.2% of included studies. Of these, eight did not provide any information on participants who were not White or referred to them as 'other'/ 'non-white'. The omission and/or inadequate description of participant ethnicity is significant in light of research demonstrating that NHS staff belonging to ethnic minorities (in the UK) may face additional stress in the workplace. The NHS staff survey [31] showed that one in five staff from minority ethnic groups (other than White minority groups) experienced discrimination. The Workforce Race Equality Standard (WRES) report showed that only 44.4% of Black and minority ethnic (BME) NHS staff felt their organisation offered equal opportunities for career progression; compared with 58.7% of White staff [129]. It is acknowledged that there is a helpful ongoing debate regarding the use of the term BME, which is not widely accepted by the groups it represents yet continues to be used. This review uses the above term only in order to refer to data from existing reports which have used it. Ethnic minority NHS staff have reported experiencing racism and racial microaggressions at work and a lack of equal opportunities in their professional roles [130]. Thus, there is substantial evidence that MHWs, along with other NHS staff, may face additional workplace stress and potentially racial trauma as a result of belonging to an ethnic minority. If organisations and researchers are not monitoring ethnic diversity when designing and implementing EBS interventions, then these are unlikely to be sufficiently inclusive or sensitive to the additional stressors faced by MHWs in minority ethnic groups.

## Theoretical underpinnings and assumptions

There was significant variation in methodology, suggesting little consensus about how research exploring EBS should be approached. Studies were grouped into three categories based on the level at which they sought to intervene, namely individual (including personal resilience; awareness-raising and role training); team and organisational approaches to addressing EBS. To contextualise these findings, these categories were considered alongside the three types of organisational interventions identified by Tetrick and Quick [43].

This comparison demonstrates all but one of the interventions were of the secondary type [43] although they were operating at various levels of individual and team intervention. The only intervention possibly acting at the organisational level and of primary type [43] was the primary nursing study by [53]. Primary nursing is a model of nursing care delivery where each patient's care is the responsibility of one nurse, with care focusing on the needs of the patient rather than the ward [131]. Therefore, this is applicable only to inpatient psychiatric settings.

Several studies provided an evidence base for the intervention rather than an explicit theoretical rationale. These studies generally noted EBS is negatively correlated with: flexibility and autonomy (flexible CBT format; primary nursing); social support and emotional reciprocity

with colleagues (peer support/relationships) and therapeutic boundaries (DBT training). Aside from Alenezi et al [59], studies utilising wellness/stress reduction interventions did not cite research specifically on EBS, instead highlighting evidence of the positive effects of these interventions on general wellbeing.

Where theoretical rationales were provided, a common thread appeared to be teaching MHWs to respond differently to the EBS inducing aspects of their jobs. For ACT, mindfulness and compassion-based interventions, this took the form of encouraging non-judgemental attitudes towards negative thoughts and feelings that might arise from work. For EBS prevention and education, the emphasis was on teaching MHWs the causes and signs of EBS so they could better respond to negative experiences. For art therapy, the emphasis was on resolving mental health issues and integrating traumatic experiences through various therapeutic art forms. PIT, supervision and some aspects of EBS-awareness (e.g. boundary setting) were the only interventions that suggest a preventative element. However, the onus was still on the MHW to change an aspect of their practice or complete an additional task. As observed by Montgomery [132], healthcare systems still dominated by the medical model take a pathogenic perspective on burnout as a problem to be 'treated' at the individual level, rather than questioning why their systems consistently produce this issue.

This individual focus aligns with a review of interventions to reduce CF in healthcare, emergency and community workers. Cocker et al [32] found all 13 included studies had an individual focus, and the majority employed stress reduction and/or holistic interventions (e.g., yoga, meditation). Several reviews of EBS interventions for healthcare staff have focused only on mindfulness-based interventions [33, 133–135]. Thus, the wider literature reflects the emphasis on individual EBS interventions as identified in this review.

However, research into factors influencing EBS suggests organisational elements are key. For example [136], demonstrated job-related factors including work environment, workload and workplace trauma were associated with CF in MHWs. One of the most consistent predictors of EBS across healthcare providers is high caseload/client contact [10, 136–138]. In a systematic review of burnout determinants [139], concluded that reasonable caseloads, clinician autonomy, good team functioning, and proper supervision should be the focus of organisational attempts to prevent and reduce burnout in MHWs. Sutton et al [35] meanwhile, found that regular supervision, balanced and diverse caseloads, strong peer support networks and an organisational culture that acknowledges secondary trauma were key to ameliorating EBS in MHWs.

Furthermore, authors have emphasised that organisations employing MHWs have a responsibility to prevent and reduce EBS [140] and warn that failure to address the systemic nature of this problem is resulting in significant clinician distress [141]. As Killian [142] noted upon finding no significant relationship between individual self-care strategies and reported levels of EBS, perhaps we should "stop expecting helping professionals to 'pull themselves up by their bootstraps' by reducing their stress with standard individual coping strategies" (pp. 42). They instead called on organisations to protect the wellbeing of MHWs by altering their workloads and giving them greater autonomy. It is possible that organisational reluctance to prioritise interventions of this nature is driven at least in part by cost-saving concerns, as meaningfully reducing workloads ultimately means hiring more staff. However, evidence suggests that improving staff engagement and satisfaction leads to better care quality, patient satisfaction, financial performance and staff retention [143, 144].

Also of note is the theory-practice gap between the known causes of EBS and the majority of interventions used to address it. STS and CF are thought to arise through traumatic exposure to the distress of others [7] and burnout via repeated interpersonal stress [6]. However, very few of the studies reviewed here addressed these issues directly. Of the seven studies

intending to target STS, only three (Boone, [Unpublished] Landis, [Unpublished]; Wymer [Unpublished]) mentioned working with trauma responses. Despite the interpersonal mechanisms implicated in EBS; most studies utilised individual interventions. The interventions that incorporated relational factors focused on relationships with colleagues and supervisors, not clients. It is therefore unclear how these interventions are proposing to address the root of the problem.

As described above, it is challenging to cohesively summarise the varied interventions outlined in this review. One lens through which to view these is the Job Demands-Resources (JD-R) model [17]. The JD-R assumes that all work consists of job demands (JDs) and job resources (JRs). JDs are effortful, psychologically or physically costly aspects of a job that are negatively valued; whilst JRs are rewarding, positively valued aspects of a job that offer personal development or mitigate the impact of JDs [145]. High JDs and insufficient JRs predict burnout. However, JDs may not increase burnout if workers also have access to resources such as autonomy and high-quality relationships with supervisors [146]. Applying this framework, most of the studies included in this review aim to increase JRs, by increasing EBS-awareness, therapeutic skill or personal resources of individual MHWs, or through team-based efforts i.e., supervision. However, decrease in JDs and increase in JRs are both necessary to enable MHWs to provide high quality, person-centred care [147].

## Measurement of EBS

There was notable inconsistency in methodology, duration of interventions and the timing/ frequency of follow ups. The Maslach Burnout Inventory was the most common measure used, consistent with the majority of studies targeting burnout. Qualitative/mixed methods were rare. There appears to have been minimal attempts to discover what factors were associated with EBS for the study populations before intervening. This may be significant given the diverse roles of MHWs samples and multiple settings featured in included studies. Bakker et al [17] describe using a two-stage process when researching the unique burnout risks in different jobs/settings. This entails qualitative interviews with workers, exploring the unique demands and resources inherent in their role. Responses are then operationalised into a custom-made questionnaire distributed to all workers. Applying this method to MHWs may be useful, and indeed there are examples to be found in the literature e.g., Wilkie et al [148] report on a two-year process of developing and implementing wellness initiatives in a Canadian hospital based on survey and consultation processes with staff.

It is beyond the remit of a scoping review to assess and compare effectiveness of interventions. However, it is worth noting that even superficial observations about the relative outcomes of different approaches were difficult to make due to: the number of different interventions used; the multi-component nature of many of the interventions, the tendency for studies to report changes across multiple subscales of EBS measures, and large variations in post-intervention follow up.

## Limitations

The inconsistent conceptualization of CF, burnout, VT and STS may hamper the usefulness of the findings to some extent. The concept of EBS [11] was applied in an attempt to overcome this challenge. Whilst this enabled an inclusive search strategy; it is possible that amalgamating concepts constituted an over-simplification. Nonetheless, over-specificity may also be unhelpful. In a review that solely targeted CF, Cocker et al [32] found their earliest study published in 2011 and concluded that the evidence base for these interventions is relatively recent. By including all EBS concepts, the current review spans 40 years of research and arguably captures

a more representative view of attempts to help MHWs manage the impact of empathically demanding work.

A further limitation was the exclusion of studies not written in English, biasing the review towards evidence produced in Western, English-speaking settings. However, this was somewhat mitigated by the search strategy, which, took a broad approach to publication status, methodology, and type of intervention.

## Research agenda

Drawing together the findings of this review with the wider literature, we have produced a research agenda to address the current issues in this field, focusing on the urgent need for effective EBS interventions in mental health services. Following the descriptions of current issues and suggested solutions, a list of summary recommendations will be provided.

### Current issues

We highlight several key issues with existing research, including:

1. Inconsistent terminology/lack of conceptual clarity, creating unhelpful partitions between similar areas of research.

2. A disconnect between the proposed causal mechanisms of EBS and interventions chosen to address it.

3. An emphasis on individually focused interventions, which is at odds with literature showing organisational factors are the leading cause of EBS.

4. An emphasis on treating the outcomes of EBS rather than addressing or preventing the causes.

5. A failure to ask MHWs themselves what the sources of EBS are in their specific organisations before implementing interventions.

6. Potentially relevant clinical audits/initiatives not being widely disseminated beyond the organisation, resulting in a disconnect between knowledge held in clinical and academic arenas.

7. Little regard for diversity factors which may intersect with EBS, resulting in poor reporting on diversity data such as ethnicity.

8. Methodological issues including inconsistency in study design/rigor, inconsistent measurement of EBS and lack of control for profession type of MHW.

### Suggested solutions

**1) Inconsistent terminology/ lack of conceptual clarity.** *Adopting and updating the EBS model.* We suggest that adopting the EBS model [11] offers a practical solution to the problem of inconsistent nomenclature in the field by distilling the core elements of the various concepts available in a dynamic process model.

The model was born out of a conceptual review and we hope that by considering it in an applied context we can support its translation into the clinical research environment. We therefore suggest one minor amendment. The current EBS model describes both individual and contextual factors influencing the onset of EBS. Examples of contextual factors are 'emotional display norms & expectations, form/frequency of trauma exposure, support' (pp. 298).

We suggest it is potentially ambiguous whether this refers to workplace or personal contextual factors. We therefore suggest a clearer distinction between organisational/job role context (e.g., workload, role autonomy, degree of exposure to EBS via work role) and individual factors (e.g., trauma history, coping style, personality, support network outside work).

*Testing the EBS model.* Empirical testing of this model is warranted e.g., identifying underlying mechanisms, determining what factors may trigger EBS, mapping how it arises over time. Additionally, determining the relative contribution of contextual and individual factors is pertinent given the emphasis on individually-focused interventions identified by this review. For example, how does the use of self-rostering systems (organisational context) to address EBS interact with caring status (individual context)?

*Differentiating concepts and updating terminology.* Following the testing processes described above, further clarification and differentiation of terms should be considered. It may be helpful to further disentangle the three empathy-based strain constructs (STS, VT, CF) from the construct of burnout (a proposed outcome of empathy-based strain- however acknowledged to share common features- [11]. Other longstanding terminology incorporated in the EBS model would benefit from review considering more recent empirical work. For example, a neuroscientific study has contested the term compassion fatigue after finding that compassion does not cause fatigue [149].

*The need for practicality and action.* Whilst consistent terminology is undeniably important when designing research and testing hypotheses, we note that the field has been plagued by conceptual disagreements for decades that are unlikely to be solved to the satisfaction of all. We therefore advocate a practical approach. It is evident MHWs can experience negative consequences from repeated empathic engagement with clients' distress and that this translates into harmful outcomes. We encourage applied, organisational and structured attempts to arrive at solutions to this chronic problem.

**2) Disconnect between the causes of EBS and the interventions used to address it.**
*Clearly linking interventions to causes.* We call for interventions that are theoretically linked to the proposed mechanisms of EBS. If studies are using individual constructs such as burnout, CF etc. then there should be clear links between the assumed mechanisms of these conditions (see Table 1) and the steps being taken to reduce or prevent them.

**3) Organisational causes vs individual solutions.** *More organisational interventions.* Existing literature suggests organisational factors are the key drivers of EBS, therefore there is a need for studies which employ organisational/primary interventions for EBS in MHWs e.g., reduced/altered caseloads, greater autonomy for staff. In their independent review into how to support nurses and midwives in delivering high quality care [150], gave primary interventions the main focus. For example, Langley Green acute mental health hospital (Sussex, UK) implemented a distributed leadership model under which staff could raise issues and implement changes to the service. In addition, supervision commitments were increased to over 90% fulfilment per week and monthly 'transparency boards' monitored levels of staff training and supervision. This model led to a decrease in staff sickness over the course of implementation and medication omissions fell from 36% to 0%. East London NHS Foundation Trust (mental health and community services) reviewed their clinical audit processes with the involvement of service users and staff and ultimately ceased 85% of all audit activity. This triggered two broader initiatives to: i) identify any activities that did not add value to services (e.g., duplication of clinical recording) which the trust then acted to address; ii) commence a 'break the rules' campaign, encouraging staff to highlight pointless procedural rules that could be eradicated. Whilst these were not research studies, they illustrate that organisational interventions do exist and would benefit from robust evaluation.

A further reason for increasing organisational interventions is suggested by the Job Demands-Resources model [17], which illustrates that job stress and burnout arise out of an imbalance between job demands and job resources. As noted above, the majority of existing studies are addressing EBS by increasing job resources rather than reducing organisational demands.

**4) An emphasis on treating the outcomes of EBS rather than addressing or preventing the causes.** *Focus on prevention*. It is easier to prevent EBS than to cure it once it has been established. Organisations employing MHWs should aim to reduce the drivers of EBS and equip themselves to recognise and intervene early when signs of EBS emerge. This focus on prevention should be reflected in the research being carried out in this field. for example, Future studies might focus on evaluating the impact of increasing MHWs autonomy, reducing excessive caseloads, self-rostering and flexi-time, higher quality and more frequent supervision, building effective teamworking and opportunities for learning, growth and development.

**5) Failure to ask MHWs about sources of EBS.** *Ask first, intervene second*. Existing research suggests there are key recurring organisational factors that can lead to EBS, with high caseloads being the most consistent, e.g. Singh et al [136]. Whilst this suggests there may be reliable trends across services, EBS-causing factors will likely vary between different organisations which differ in terms of structure, remit and funding. When developing interventions to address EBS in MHWs, a suggested preliminary step is for researchers to consult with MHWs to gain an understanding of the sources/causes of EBS in their particular organisation/setting, and design interventions accordingly. The two-step process described by Bakker et al [17] (see discussion) represents a further example of using specific staff concerns to generate outcome measures for gauging the success of an intervention.

**6) Disconnect between clinical and academic domains.** *Practice-informed research*. The in-house clinical initiatives and audits described by West et al [144] demonstrate that organisational and primary interventions are happening. However, such data is not typically accessible to researchers. We therefore recommend greater connection and collaboration between clinical and academic organisations seeking to tackle the issue of EBS. This would benefit both the clinical staff and services participating in studies (e.g., greater chance of having context specific and relevant issues addressed); the research teams themselves (e.g., help to identify meaningful targets for outcome monitoring; ideas for intervention) and the quality and usefulness of the findings (e.g., greater validity and relevance).

**7) Insufficient attention given to diversity factors.** *Record and control for diversity*. Researchers must attend to and address the additional stressors that may be faced by MHWs who belong to minority groups when designing interventions for EBS e.g., by using additional measures that capture experiences of discrimination.

**8) Methodological issues.** *Higher quality, consistent methodology*. Most of the studies included in this review made recommendations for future research (see S1 File). The most consistent of these was for greater methodological rigor i.e., controlled, experimental designs and larger samples. In order for future reviews to draw conclusions about the relative effectiveness of different EBS interventions, there is a need for greater methodological consistency across studies.

*Improving outcome measurement*. The second most common recommendation from included papers was for future studies to have a longer duration of intervention and/or follow up prior to measurement. Several studies also recommended measuring whether targeting EBS in MHWs leads to improved outcomes for clients. Greater consistency in measurement tools would also allow direct comparison of studies.

*Controlling for specific MHWs role*. Several studies in the review acknowledged the diverse professions and settings that MHWs may represent and suggested this be controlled for in future studies.

## Summary recommendations

For ease of reference, we have summarised the above into bullet points intended to guide researchers undertaking further studies into interventions for EBS in MHWs. Future studies should:

- Prioritise applied intervention research over conceptual debate.

- Consider adopting the EBS model [11] to overcome conceptual discrepancies.

- Carry out empirical testing of the EBS framework and clarify terminology accordingly.

- Ensure that interventions are theoretically linked with the proposed causes of EBS.

- Prioritise organisational or primary interventions (e.g., caseloads, service structure, worker autonomy) over those that target individual MHW qualities or skills.

- Design studies that reduce job demands rather than solely attempting to provide job resources.

- Prioritise studies that focus on preventing EBS rather than seeking to cure it once it has been established.

- Find out from MHWs what the sources of EBS are in their particular setting before intervening.

- Collaborate with services and organisations employing MHWs to share knowledge and data between the clinical and academic realms.

- Measure and control for diversity factors in participants, to ensure interventions are applicable to diverse MHWs.

- Design larger, more methodologically robust experimental studies that control for the setting and specific role of MHWs.

- Consider longer periods of intervention and follow up prior to/over the course of outcome measurement.

- Consistency in the EBS measures used to allow better comparison between studies.

- Consider measuring whether the EBS interventions for MHWs improve outcomes for their clients.

## Conclusion

This scoping review outlined the current literature regarding interventions for EBS in MHWs. Overall, the findings revealed little consistency across available interventions; aside from a tendency to intervene at the level of the individual. We have questioned whether this individual focus is helpful or sufficient considering the organisational drivers of EBS. Finally, we have presented a research agenda detailing how these and other issues can be addressed by future research.

## Supporting information

**S1 File. Supporting information.**
(DOCX)

## Author Contributions

**Conceptualization:** Hannah May, Emma Griffith.

**Data curation:** Hannah May, Mhairi Kristoffersen, Ross Robinson.

**Formal analysis:** Hannah May.

**Methodology:** Josie Millar.

**Supervision:** Josie Millar, Emma Griffith, Chris Gillmore, Michael West.

**Writing – original draft:** Hannah May.

**Writing – review & editing:** Hannah May, Josie Millar, Emma Griffith, Chris Gillmore, Michael West.

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
