## [Decision Letter · Decision Letter 0]

18 Dec 2023

PONE-D-23-33814Interventions to address empathy-based stress in mental health workers: A scoping review and research agendaPLOS ONE

Dear Dr. May,

Thank you for submitting your manuscript to PLOS ONE. After careful consideration, we feel that it has merit but does not fully meet PLOS ONE’s publication criteria as it currently stands. Therefore, we invite you to submit a revised version of the manuscript that addresses the points raised during the review process.

**ACADEMIC EDITOR Feedback**Dear Authors,

We greatly appreciate the efforts put into the paper entitled "Interventions to Address Empathy-Based Stress in Mental Health Workers: A Scoping Review and Research Agenda." Following the comprehensive review by our esteemed peer reviewers, we've collated their feedback and suggestions to enhance the quality and impact of your work: in addition to the deialed reports of the reviewers .consider addressing all of the following:

Overall Paper Structure and Content:

1. Title and Abstract:Clarify that the paper is a scoping review rather than focusing on interventions directly. Ensure the abstract introduces the four main constructs and succinctly highlights the research's unique contributions.

2. Introduction: Condense the introduction, starting from line 54, and emphasize the gap in knowledge the research addresses. Create a clearer link between the conceptual model and mental health workers (MHW).

3. Methods:Enhance clarity on scoping review protocols, from identifying the research question to reporting results. Explicitly state the research question and protocol used.

4.Results:Revise Table 4 to cover all variables of interest. Visualize characteristics of included studies in percentages and incorporate trend analysis over the past 20 years.

5. Intervention: Streamline this section, focusing on the most impactful interventions. Provide clarity on how these interventions align with the research questions and proposed protocol. Balance coverage of different interventions.

6. Discussion: Condense the discussion for conciseness. Also, ensure the paper undergoes thorough English editing.

7. References: Update and expand the reference list to cover a wider time span and ensure conformity to the journal's referencing style.

Addressing Specific Reviewer Comments: in addition to deataled comments of reviewer 1. consider the following

1. Reviewer 2: Clarify the study's unique aspects compared to previous research. Ensure consistency in referencing style.

2. Reviewer 3: Enhance the readability of the research agenda section.

3. Reviewer 4: Address the conflict between healthcare organizations and providers, emphasizing structural interventions. Consider the U.S. healthcare perspective and its impact on the paper's content.

4. Reviewer 5: Organize the content for better flow, specifically around outlining intervention levels early, providing clear linkages between findings and sources, and expanding on prevention-focused research efforts.

General Recommendations:

Clarity and Conciseness: Ensure each section is concise, coherent, and directly addresses the objectives and contributions of the paper.

Consistency and Detail: Maintain consistency in referencing style and detail the unique contributions of the study compared to existing literature.

Addressing Gaps: Highlight the gap in knowledge that the paper addresses and ensure the conceptual model directly relates to mental health workers.

Visual Representation: Use tables and figures effectively to illustrate key points, characteristics of studies, and trends in the field.

Addressing these areas should significantly improve the paper's quality and address the concerns raised by the reviewers. It may require restructuring, condensing, and enhancing clarity throughout the document. Additionally, consider seeking professional assistance for language editing to ensure grammatical correctness and overall readability.

We look forward to receiving your revised manuscript.

Kind regards,

Prof. Ebtsam Abou Hashish, 

Academic Editor

PLOS ONE

3. PLOS requires an ORCID iD for the corresponding author in Editorial Manager on papers submitted after December 6th, 2016. Please ensure that you have an ORCID iD and that it is validated in Editorial Manager. To do this, go to ‘Update my Information’ (in the upper left-hand corner of the main menu), and click on the Fetch/Validate link next to the ORCID field. This will take you to the ORCID site and allow you to create a new iD or authenticate a pre-existing iD in Editorial Manager. Please see the following video for instructions on linking an ORCID iD to your Editorial Manager account: https://www.youtube.com/watch?v=_xcclfuvtxQ.

4. We note that you have referenced (Ballew, J.K. (2020) The effects of resiliency training on self-reported compassion fatigue and compassion satisfaction in mental health professionals and counselors-in-training), (Boone, B.C. (2012). The impact of poetry therapy on symptoms of secondary posttraumatic stress disorder in domestic violence counsellors), (Boone, B.C. (2012). The impact of poetry therapy on symptoms of secondary posttraumatic stress disorder in domestic violence counsellors), (Gentry, J. E. (1996). Solution-focused trauma recovery scale (TRS), (Landis. E.M. (2010). Sharevision: A Collarative-Reflective, Expressive Arts Intervention To Address Trauma), ( Ray, C.A. (1981). Holistic stress management training: a burnout strategy for mental health workers), (Ray, C.A. (1981). Holistic stress management training: a burnout strategy for mental health workers), (Walker, D. (2018). Examining the Effects of a Brief Mindfulness-Based Stress Reduction Program on Burnout in Mental Health Professionals), and (Wymer, B. (2019). An Investigation of the Impact of a Supervision Intervention on Secondary Traumatic Stress Responses Among Counselors Treating Child Survivors of Sexual Abuse) which have currently not yet been accepted for publication. Please remove this from your References and amend this to state in the body of your manuscript: (ie “Bewick et al. [Unpublished]”) as detailed online in our guide for authors

5. Please remove your figure 2 from within your manuscript file, leaving only the individual TIFF/EPS image files, uploaded separately. These will be automatically included in the reviewers’ PDF.

6. Please include a caption for figures 1 and 3.

7. We notice that your supplementary files are included in the manuscript file. Please remove them and upload them with the file type 'Supporting Information'. Please ensure that each Supporting Information file has a legend listed in the manuscript after the references list.

Reviewers' comments:

Reviewer's Responses to Questions

**Comments to the Author**

1. Is the manuscript technically sound, and do the data support the conclusions?

Reviewer #1: Partly

Reviewer #2: Yes

Reviewer #3: Yes

Reviewer #4: Yes

Reviewer #5: Partly

2. Has the statistical analysis been performed appropriately and rigorously? 

Reviewer #1: I Don't Know

Reviewer #2: N/A

Reviewer #3: N/A

Reviewer #4: N/A

Reviewer #5: I Don't Know

3. Have the authors made all data underlying the findings in their manuscript fully available?

Reviewer #1: No

Reviewer #2: Yes

Reviewer #3: Yes

Reviewer #4: Yes

Reviewer #5: Yes

4. Is the manuscript presented in an intelligible fashion and written in standard English?

Reviewer #1: No

Reviewer #2: Yes

Reviewer #3: Yes

Reviewer #4: Yes

Reviewer #5: Yes

5. Review Comments to the Author

Reviewer #1: Thank you for giving me this chance to revise this valuable paper with valuable variable that can affect healthcare worker specially nurses and physician and can lead to increase turnover

The title of the research is not clear , it is not intervention it is scoping reviews

The abstract :too long , the four construct must be showed in introduction of abstract

Introduction

• what already known and what is added by your research

• very long you can start from the line no( 54) and summarize the paragraph start with line no:77-89

• figure no 1 not included in the paper.

• the other construct of the conceptual model need to be related to MHW as CF(line no 102:112)

• the significant of the study need to be clear and concise it was scattered in the introduction.

• the gab of knowledge /this research need to be identified.

• The conceptual frame work need to be more clear and identify each construct with adequate references

Methods

• Scoping reviews tend to focus on the nature, volume, or characteristics of studies rather than on the synthesis of published data. In health care it would prefere to uses systemic to decrease bias and chose best in class research.

• The research question need to be added

• The protocol of scoping reviews need to be addeding

• The steps of scope reviews must be covered from identifying research question till Collating, Summarizing, and Reporting the Results with clear explanation of each step

• Results :

• In table 4 the scope review not include each of VC also more than 80% of research concentrate on burnout although the research conclude in the introduction it is results more than construct.

• Characteristics of Included Studies need to be visualized in no & percent

• Trend analysis used to present the changing frequency of research over the past 20 years, based on the aforementioned classification criteria need to be added

Intervention

• This part are very long need to be shorten and focus on the most affected intervention that can decrease EBS or burnout.

• The intervention must be declared to what extent it support or against or answer the authors research questions and cover the proposed protocol

• The balance between different interventions was lost in this section . it focuses only on three interventions while the other intervention reference was very little

Discussion

• It isy long and need to be concise

• Although I am not native speakers but the paper as all need to be review by English editors

References :

• More than 50 refrence rang rom 1953 to2009

Reviewer #2: Thank you for the opportunity to read this interesting and detailed article. I provide my comments/suggestions which the authors may find useful for improving the overall quality of the paper.

1. I think the use of questions and tables to introduce this article does not make the Introduction stronger. Authors may reorganize this section. In its current form is more like giving out information to readers, rather than presenting a problem, telling what has been done about it, and what needs to be done which you are going to do.

2. In terms of the study justification, I think the authors may go deeper. For instance, one of the cited studies ( Bercier and Maynard, 2014) to justify the current study is not even in the reference list for confirming what has been stated in the article. Also, the authors state that they included studies with at least 50% of the participants being mental health workers. How is their study different from earlier studies (such as Conversano et al., 2020) that combined mental health workers and other participants?

3. Authors may present the study objectives more concisely.

4. Same with the methods, especially from line 146 to 167. This could be reorganized to make it more concise.

5. The in-text citations are not conforming to the journal’s referencing style. Also, some of the in-text citations cannot be found in the reference list. An important example is Bercier and Maynard (2014) cited in line 117 to justify the need for the present study.

6. Authors may be consistent with the use of "Whilst" and "While"

Reviewer #3: Thanks for an interesting and informative review. Would be helpful to detail the data synthesis done and if possible make the research agenda appendix to the review to reduce length and aid readability. Other comments attached in the reviewed document

Reviewer #4: Thank you very much for the opportunity to review this manuscript on a scoping review and research agenda regarding interventions that target empathy-based stress (EBS) in mental health workers (MHWs). I found the review steps and presented information to be very comprehensive and clear, and I particularly appreciated the authors’ (i) acknowledgement of categorizations of findings not necessarily being mutually exclusive (e.g., Lines 340-342) and (ii) descriptions of specific inclusion/exclusion decisions that required additional thought (e.g., 328-339). My suggestions below are mostly to enhance (i) flow and clarity of the manuscript’s covered concepts and (ii) linkages between the reported findings and the articles from which they originate.

1. Please consider moving the information in Lines 136-144 (which do not seem to be MHW-specific) to be earlier in the Introduction section, perhaps around Lines 54-55 that describe why EBS deserves attention (i.e., before Line 90, which starts to describe the rationale for studying EBS in MHWs specifically).

2. Please consider outlining the three intervention levels of Tetrick and Quick (2011) when they are first mentioned in the manuscript (Lines 231-232), rather than later on in Lines 384-392.

3. Please consider previewing around Lines 647-650 that, following the descriptions of current issues and suggested solutions in the Research Agenda section, a list of summary recommendations will be provided.

4. Please consider making clearer which exact reviewed articles are associated with each different type of (i) broad categories of interventions, (ii) routes of intervention, (iii) measures, and (iv) recommendations for future research, which the manuscript describes very usefully in Table 6, the “Route of Intervention” section (starting on Line 365 and Figure 3), Appendix C, and Appendix D, respectively.

5. Please consider expanding, around Lines 754-758, on some specific examples of potential ways in which future research efforts can reflect a focus on prevention.

6. Please consider clarifying, throughout the manuscript where the search dates are mentioned, that the updated search included articles up until September 2022 (since Line 183 mentions that the updated search was conducted in September 2022), rather than just noting “2022,” which indicates that articles from all months of 2022 were included in the review.

Reviewer #5: This paper reflects an enormous amount of work that summarizes interventions to address empathy-based stress (EBS) in mental health care workers. It includes the definitions of EBS and the details of the interventions that have been studied to reduce it. This summary will be of value to future researchers.

I believe that the paper's main shortcoming is its failure to address the inherent conflict between health care organizations and the health care providers they employ. For example, in the U.S. there are numerous newspaper articles about health care organizations seeking to hire as few staff as they can get away with and health care workers seeking to unionize and/or going on strike to assert their rights to reasonable work conditions.

It's naive to assume health care organizations are motivated to make such structural interventions such as reducing workload, even if in the long run it would benefit them to do so. This paper needs to address conflicts between the interests of health care workers and the economic drivers that cause health care organizations to deliver care as inexpensively as possible.

6. PLOS authors have the option to publish the peer review history of their article (what does this mean?). If published, this will include your full peer review and any attached files.

Reviewer #1: No

Reviewer #2: No

Reviewer #3: **Yes: **Dr. Clement Nhunzvi

Reviewer #4: **Yes: **Bo Kim

Reviewer #5: **Yes: **Francine Cournos, M.D. Professor of Clinical Psychiatry (in Epidemiology), Columbia University, New York, NY

---

## [Author Response · Author response to Decision Letter 0]

18 Mar 2024

ACADEMIC EDITOR Feedback

Overall Paper Structure and Content:

1. Title and Abstract: Clarify that the paper is a scoping review rather than focusing on interventions directly. Ensure the abstract introduces the four main constructs and succinctly highlights the research's unique contributions.

Response: Thank you for this comment. We have amended the abstract (pg 2) to more clearly identify the paper as a scoping review and to emphasise its unique contribution. The paper is also identified as a scoping review within its title. We have also added reference to the four main constructs of EBS to the abstract.

2. Introduction: Condense the introduction, starting from line 54, and emphasize the gap in knowledge the research addresses. Create a clearer link between the conceptual model and mental health workers (MHW).

Response: Thank you for this comment. We have amended the introduction to more clearly emphasise the relevance of the EBS model to mental health workers (pg 4). We have also condensed the introduction as requested (pgs 2-8). 

3. Methods: Enhance clarity on scoping review protocols, from identifying the research question to reporting results. Explicitly state the research question and protocol used.

Response: Thank you for this comment. The research question and associated aims are given in the ‘Current Review and Aims’ section (pg 7), and copied below for ease of reference:

““What interventions exist to prevent or treat EBS in MHWs?”. Using systematic scoping methodology, we aimed to answer this question via the following objectives:

1) To assess and summarise the available interventions for preventing or treating EBS in MHWs

2) To ascertain the theoretical underpinnings and assumptions of these interventions

3) To assess how EBS has been measured in these studies

4) To make recommendations for future research into the treatment and prevention of EBS in MHWs”

As described in the methods section, the review was conducted in accordance with the Joanna Briggs Scoping Review Framework (Peters et al., 2017) and written up in line with the Prisma Extension for Scoping Reviews (Tricco et al., 2018). As described in line 168, An a priori protocol was developed and registered on Open Science Framework and is available at the following link: https://osf.io/b7kcr/

4.Results: Revise Table 4 to cover all variables of interest. Visualize characteristics of included studies in percentages and incorporate trend analysis over the past 20 years.

Response: Thank you for this suggestion. We have reported the characteristics of the included studies in accordance with the Joanna Briggs Scoping Review Framework (Peters et al., 2017) and the Prisma Extension for Scoping Reviews (Tricco et al., 2018) In line with these, Table 4 of the scoping review is not intended to provide further analysis but to present the characteristics of included studies which are relevant to the research question. Table 4 in the manuscript accordingly presents this data. The distribution of included studies by year of publication is not relevant to our research questions. For example, the Joanna Briggs Scoping Review Framework (Peters et al., 2017) instructs:

“In a scoping review, the results may be presented as a ‘map’ of the data in a logical, diagrammatic, or tabular form, and/or in a descriptive format that aligns to the objective/s and scope of the review. The tables and charts may show results as: distribution of studies by year or period of publication (depends on each case), country of origin, area of intervention (clinical, policy, educational, etc.), and research methods. It is up to the reviewers to decide which would most rationally and clearly illustrate the nature of the results in terms of the objective/s and question/s of the review. A summary of the results should logically describe the aims or purposes of the included articles, the concepts or approaches adopted in each, and the results that relate to the review question/s.”

5. Intervention: Streamline this section, focusing on the most impactful interventions. Provide clarity on how these interventions align with the research questions and proposed protocol. Balance coverage of different interventions.

Response: Thank you for this comment. The section reporting on interventions to reduce/prevent EBS (starting on pg 28) is part of the results section, and is therefore aligned with the first aim of the research question: “ To assess and summarise the available interventions for preventing or treating EBS in MHWs”.

We have retitled this subsection of the results section “Summary of Available Interventions for Preventing or Treating EBS in MHWs” to more clearly orient the reader to the purpose of this section and link it back to the relevant research aim. 

 We have made edits (pg 28-39) in order to enhance clarity and readability of this section. However, identifying the most impactful interventions was not one of our aims, nor is it compatible with the purpose of a scoping review. As pointed out on pg 46: 

“It is beyond the remit of a scoping review to assess and compare effectiveness of interventions. However, it is worth noting that even superficial observations about the relative outcomes of different approaches were difficult to make due to: the number of different interventions used; the multi-component nature of many of the interventions and the tendency for studies to report changes across multiple subscales of EBS measures.”

6. Discussion: Condense the discussion for conciseness. Also, ensure the paper undergoes thorough English editing.

Response: thank you for this comment. We have condensed the discussion for conciseness (pgs 39-46) and reviewed English editing for the whole manuscript. 

7. References: Update and expand the reference list to cover a wider time span and ensure conformity to the journal's referencing style.

Response: Thank you for this comment. As described on pg 9, the time span of the review was based on the history of the relevant constructs. As identified by Schaufeli et al (2009), the term burnout (the oldest of the constructs) was first used in 1970 and therefore this was the starting range for searches. We have reviewed the referencing and ensured conformity to the journals referencing style. 

Addressing Specific Reviewer Comments: in addition to detailed comments of reviewer 1. consider the following

1. Reviewer 2: Clarify the study's unique aspects compared to previous research. Ensure consistency in referencing style.

Response: thank you for this comment. We have reviewed the introduction in line with the comments regarding the narrative as suggested here, and also have edited for clarity and conciseness throughout as suggested by multiple reviewers. The references have been reviewed for completeness and for adherence to the journal’s style. See responses to Reviewer 2 below for more detail. 

2. Reviewer 3: Enhance the readability of the research agenda section. 

Response: thank you for this comment. The research agenda has been edited for readability 

3. Reviewer 4: Address the conflict between healthcare organizations and providers, emphasizing structural interventions. Consider the U.S. healthcare perspective and its impact on the paper's content.

Response: thank you for this comment. It is an interesting observation, however, the available evidence suggests that improving staff engagement and satisfaction leads to precisely the outcomes healthcare organisations seek. Data from the largest annual staff survey in healthcare show that improving engagement and satisfaction leads to better care quality, patient satisfaction, financial performance and staff retention. We have added this information, with references, to page 6.

4. Reviewer 5: Organize the content for better flow, specifically around outlining intervention levels early, providing clear linkages between findings and sources, and expanding on prevention-focused research efforts.

Response: Thank you for this comment. We have edited for better flow, outlined intervention levels by Tetrick & Quick where they first appear and expanded on prevention-focused research ideas as suggested. Please see individual responses below for further details. 

General Recommendations:

Clarity and Conciseness: Ensure each section is concise, coherent, and directly addresses the objectives and contributions of the paper.

Response: Thank you for this comment. We have edited the paper for conciseness and clarity based on feedback from multiple reviewers.

Consistency and Detail: Maintain consistency in referencing style and detail the unique contributions of the study compared to existing literature.

Response: Thank you for this comment. As noted above, we have amended lines the abstract (pg 2) to more clearly identify the paper as a scoping review and to emphasise its unique contribution. We have also corrected referencing style to be in line with journal style.

Addressing Gaps: Highlight the gap in knowledge that the paper addresses and ensure the conceptual model directly relates to mental health workers.

Response: Thank you for this comment. As noted above, we have amended the abstract (pg 2) to more clearly identify the paper as a scoping review and to emphasise its unique contribution. We have also amended the introduction more clearly emphasise the relevance of the EBS model to mental health workers (pg 4). 

Visual Representation: Use tables and figures effectively to illustrate key points, characteristics of studies, and trends in the field.

Response: Thank you for this comment. As noted above, we have not produced a visual trend analysis as we have reported the characteristics of the included studies in accordance with guidance on conducting and reporting systematic scoping reviews. In line with these, the results section of the scoping review is not intended to provide further analysis but to present the characteristics of included studies which are relevant to the research question. Table 4 in the manuscript accordingly presents this data. The distribution of included studies by year of publication is not relevant to our research questions. For example, the Joanna Briggs Scoping Review Framework (Peters et al., 2017) instructs:

“In a scoping review, the results may be presented as a ‘map’ of the data in a logical, diagrammatic, or tabular form, and/or in a descriptive format that aligns to the objective/s and scope of the review. The tables and charts may show results as: distribution of studies by year or period of publication (depends on each case), country of origin, area of intervention (clinical, policy, educational, etc.), and research methods. It is up to the reviewers to decide which would most rationally and clearly illustrate the nature of the results in terms of the objective/s and question/s of the review. A summary of the results should logically describe the aims or purposes of the included articles, the concepts or approaches adopted in each, and the results that relate to the review question/s.”

Addressing these areas should significantly improve the paper's quality and address the concerns raised by the reviewers. It may require restructuring, condensing, and enhancing clarity throughout the document. Additionally, consider seeking professional assistance for language editing to ensure grammatical correctness and overall readability.

Response: Thank you for this comment. The manuscript has been reviewed for grammatical accuracy. 

Reviewer 1

The title of the research is not clear , it is not intervention it is scoping reviews

Response: Thank you for this comment. We have amended lines the abstract (pg 2) to more clearly identify the paper as a scoping review and to emphasise its unique contribution. The paper is also identified as a scoping review within its title. 

The abstract :too long , the four construct must be showed in introduction of abstract

Response: Thank you for this comment. We have included reference to the four constructs of EBS within the abstract (page 2). PLOSONE journal submission guidance states that abstracts should not exceed 300 words. Our abstract is 228 words, including both this addition and minor additions requested by reviewers in relation to separate comments. 

Introduction 

• what already known and what is added by your research

Response: Thank you for this comment. Thie aim of this reviews were, as summarized in the ‘current review and aims section’ (pg 7):

1) “To assess and summarise the available interventions for preventing or treating EBS in MHWs

2) To ascertain the theoretical underpinnings and assumptions of these interventions

3) To assess how EBS has been measured in these studies

4) To make recommendations for future research into the treatment and prevention of EBS in MHWs”

 Therefore, the knowledge that is added is in synthesizing the findings of the scoping review . The outcome of which is reported on in the results section and elaborated on in the discussion. 

• very long you can start from the line no( 54) and summarize the paragraph start with line no:77-89

Response: Thank you for this comment. We have edited the paper for conciseness and clarity throughout based on feedback from multiple reviewers, with particular focus on the introduction.

• figure no 1 not included in the paper. 

Response: thank you for this comment. Figure 1 as uploaded in the submission is a diagram showing the EBS model, reproduced with permission by the authors of the Rauvola et al (2019) paper where this first appeared. The traditional ‘figure 1’ i.e. the PRISMA flow chart is included as Figure 2. 

• the other construct of the conceptual model need to be related to MHW as CF(line no 102:112)

Response: Thank you for this comment. We are unsure what exactly is being requested here. The review (Turgoose & Maddox, 2017) referenced in the section indicated by the reviewer does specifically reference compassion fatigue as an example, however the subsequent references refer to EBS more broadly. In addition, whilst we have made reference to individual constructs (compassion fatigue, burnout etc) where appropriate to the specific paper being cited, the purpose of using the umbrella term EBS is to highlight the high degree of conceptual overlap and inter-relatedness of these concepts. The rationale for doing this and the EBS model is described in the introduction. 

• the significant of the study need to be clear and concise it was scattered in the introduction.

• the gab of knowledge /this research need to be identified.

Response: Thank you for these two comments, which we are responding to as one as they seem to make overlapping points. As above, we have edited the paper for conciseness and clarity with particular focus on the introduction. We would also highlight that the “Current Review and Aims” (pg 7) summarises the unique contribution of the review and the gap of knowledge it is addressing. 

• The conceptual frame work need to be more clear and identify each construct with adequate references

Response: Thank you for this comment, details of each construct along with references are outlined in Table 1. 

 Methods

• Scoping reviews tend to focus on the nature, volume, or characteristics of studies rather than on the synthesis of published data. In health care it would prefere to uses systemic to decrease bias and chose best in class research. 

Response: Thank you for this comment. Our review question (pg 7) was “What interventions exist to prevent or treat EBS in MHWs?”. Using systematic scoping methodology, we aimed to answer this question via the following objectives:

1. To assess and summarise the available interventions for preventing or treating EBS in MHWs

2. To ascertain the theoretical underpinnings and assumptions of these interventions

3. To assess how EBS has been measured in these studies

4. To make recommendations for future research into the treatment and prevention of EBS in MHWs”

In accordance with these aims, a scoping methodology was the most appropriate as our intention was not to assess relative effectiveness of interventions but to understand what interventions were being used, the theoretical reationale for them and how EBS was measured. 

---

## [Decision Letter · Decision Letter 1]

9 Apr 2024

PONE-D-23-33814R1Interventions to address empathy-based stress in mental health workers: A scoping review and research agendaPLOS ONE

Dear Dr. May,

Thank you for submitting your manuscript to PLOS ONE. After careful consideration, we feel that it has merit but needs further revision. Therefore, we invite you to submit a revised version of the manuscript that addresses the points raised during the review process.. Based on the feedback from Reviewer 1 and Reviewer 5, it is evident that there are several areas of improvement needed for the manuscript. The reviewers have highlighted issues related to clarity, organization, referencing, conciseness, and addressing conflicts within the healthcare system.I recommend considering the following modifications cited below

We look forward to receiving your revised manuscript.

Kind regards,

Ebtsam Aly Omer Abou Hashish

Academic Editor

PLOS ONE

Journal Requirements:

Additional Editor Comments:

Dear authors

Thanks for your revised manuscript before proceeding to the final decision. I recommend considering the following modifications: 

The title must include scoping review, as it indicates the type of your papers.

The abstract: The introductory paragraph needs to be more concise and focused;

The conclusion of the study must be written, as must the implications for healthcare organizations. Please find the reference number on page 4, lines 69–72. Introduction :

The background of the topic needs to be clarified.

Introduction from pages 2–5 You need to be more concise, as it is very long, to address the key concept of your research.

The words clinician on pages 5 and 6 need to be more categorized into nurses, physicians, and so on.

The review and aim: The current related literature needs to be supported by other studies in different settings, countries, and sectors (private and governmental).

The aim is Objective No. 5 is very general and cannot be considered an aim as it is an essential part of the research paper.

The letter t on page 7, line 131, needs to be deleted, as well as repeated (.) in line 135.

The result needs to be more concise, as it is very long.

Discussion: It needs to be organized according to the most important intervention, as this paper aims to clarify all interventions and identify the most applicable in healthcare.

Recommendations need to be concise, and only applicable recommendations

The paper is very long and needs to be more concise. The authors should choose the most applicable intervention to prevent confusion with the readers.

Consider the previous reviewer comment about the potential conflicts between the needs of health care workers and the needs of the leadership of health care organizations. The authors have stated, "Most studies intervened at the level of the individual, despite the proposed causes of EBS being predominantly organizational." The authors don't address why this is happening. The authors point out that excessive workload is a major contributor to EBS, yet there were no interventions to reduce workload by hiring more healthcare workers. In most circumstances, there are in fact more healthcare workers available to be hired, but doing so is costly. That's the obvious conflict the authors need to address. Organizations usually try to get by with as few workers as possible to save costs. Workers then pay the price by having excessive workloads. If the authors have a different explanation for why organizations don't correct this obvious situation, they should offer it.

Reviewers' comments:

Reviewer's Responses to Questions

**Comments to the Author**

1. If the authors have adequately addressed your comments raised in a previous round of review and you feel that this manuscript is now acceptable for publication, you may indicate that here to bypass the “Comments to the Author” section, enter your conflict of interest statement in the “Confidential to Editor” section, and submit your "Accept" recommendation.

Reviewer #1: (No Response)

Reviewer #5: All comments have been addressed

2. Is the manuscript technically sound, and do the data support the conclusions?

Reviewer #1: Yes

Reviewer #5: Yes

3. Has the statistical analysis been performed appropriately and rigorously? 

Reviewer #1: N/A

Reviewer #5: I Don't Know

4. Have the authors made all data underlying the findings in their manuscript fully available?

Reviewer #1: Yes

Reviewer #5: Yes

5. Is the manuscript presented in an intelligible fashion and written in standard English?

Reviewer #1: Yes

Reviewer #5: Yes

6. Review Comments to the Author

Reviewer #1: I congratulate you for this work you have carried out to fill the gap in the literature. I think that this article will be a precursor for future studies. In addition, the results obtained can contribute to institutional managers taking measures to solve problems, supporting employees and creating awareness of empathy based stress among mental health workers , which is a very important issue.

Title must include scoping review as it indicate the type of your papers

The abstract:

The introductory paragraph need to be more concise and focused, the conclusion of the study must be written and the implication in healthcare organizations

Please rite the reference no in page 4 line 69,72

Introduction :

The background of the topic need to be more clarified

Introduction from page 2-5 Need to be more concise as it is very long, to address the key concept of your research

The word clinician in page 5, and 6 need to be more categorized into nurses, physicians an so on

The revies and aim:

The current related literature need to supported by other studies in different setting and countries and sectors(private, governmental)

The aim : objectives no 5 is very general and cannot be consider aim as it is essential part of the research paper

Letter t in page 7 line 131 need to be deleted also repeated (.) in line 135

The result need to more concise as it is very long

Discussion : need to be organized according to the most important intervention as this paper aims to clarify all intervention and identify the most applicable on in healthcare.

Recommendation need to be concise and choose only applicable recommendations

Reviewer #5: I don't think the authors understood my previous comment about potential conflicts between the needs of health care workers and the needs of the leadership of health care organizations. The authors have stated "Most studies intervened at the level of the individual, despite the proposed causes of EBS being predominantly organizational." The authors don't address why this is happening. The authors point out that excessive workload is a major contributor to EBS, yet there were no interventions to reduce workload by hiring more healthcare workers. In most circumstances there are in fact more healthcare workers available to be hired, but doing so is costly.. That's the obvious conflict the authors need to address. Organizations usually try to get by with as few workers as possible to save costs. Workers then pay the price by having excessive workloads. If the authors have a different explanation for why organizations don't correct this obvious situation, they should offer it.

7. PLOS authors have the option to publish the peer review history of their article (what does this mean?). If published, this will include your full peer review and any attached files.

Reviewer #1: No

Reviewer #5: **Yes: **Francine Cournos, M.D.

---

## [Author Response · Author response to Decision Letter 1]

18 May 2024

The below response is also included in the uploaded document 'Cover Letter/Response to Reviewers.

Journal Requirements:

Response: thank you for raising this important consideration. We can confirm that the original source for each reference cited (unless otherwise specifically stated in the manuscript) was sighted and reviewed by the authors during the process of undertaking the scoping review. Thus, we can confirm that at the point in time that each reference was cited/ added to the manuscript it was complete and accurate, and there was no issue relating to the publication status of any references. To the best of the authors knowledge this continues to be the case and thus we confirm that we have not knowingly cited any papers that have been retracted. Following this comment we have used an AI tool and the reference check function on Scite (https://scite.ai/home) to check the reference list for any retractions or known issues with cited papers, neither of which flagged any references of concern.

However, If the reviewer raising this comment, is aware of a cited paper that has been retracted, we would be grateful if they could please alert us of the specific reference. We will then of course take the steps as they have described to address and amend this. 

Additional Reviewer Comments:

2. The title must include scoping review, as it indicates the type of your papers.

Response: thank you for this comment. Please note the words ‘scoping review’ are included in the title, Title: “Interventions to address empathy-based stress in mental health workers: A scoping review and research agenda”.

3. The abstract: The introductory paragraph needs to be more concise and focused;

The conclusion of the study must be written, as must the implications for healthcare organizations. Please find the reference number on page 4, lines 69–72. 

Response: thank you for this comment. We have now made edits to the beginning of the abstract to make it more concise. We have also included a conclusion and implications for healthcare organisations. Regarding the lines quoted, the positioning of reference has been edited for clarity. 

4. Introduction :

The background of the topic needs to be clarified.

Response: thank you for drawing out attention to this. In order to clarify the background to the topic, we have outlined the context for the review which provides the context for why this is an important issue, what has been done and the existing gaps. We have been mindful to balance this in line with requested amendment no 5 (as per below) and thus have edited the introduction on the pages suggested to ensure it is more concise. 

5. Introduction from pages 2–5 You need to be more concise, as it is very long, to address the key concept of your research.

Response: thank you for this feedback. We appreciate the introduction is lengthy and we have grappled with the challenge of summarising the large amount of information required to adequately contextualise the review. However, we have now made further edits to pages 2-5 to reduce length and increase clarity.

6. The words clinician on pages 5 and 6 need to be more categorized into nurses, physicians, and so on.

Response: thank you for this comment. We have changed the word clinician to be more specific as requested. 

7. The review and aim: The current related literature needs to be supported by other studies in different settings, countries, and sectors (private and governmental).

Response: thank you for this comment. We have utilised a range of literature from different settings, countries, and sectors. This includes the National Health Service (government run) and academic literature from studies conducted in a range of countries ( across Europe, North America, Asia and Africa) and settings both government and private sector. 

In line with the aims of the review we have drawn on data from across healthcare fields that have focussed on mental health care settings. Please note that included studies (though all were written in English due to this being a scoping review) were drawn from 10 separate countries and from child and adult MH settings including inpatient, outpatient/community, education, forensics, veterans services, addictions service and domestic violence services.

8. The aim is Objective No. 5 is very general and cannot be considered an aim as it is an essential part of the research paper.

Response: thank you for this comment, we have removed this from the manuscript. 

9. The letter t on page 7, line 131, needs to be deleted, as well as repeated (.) in line 135.

Response: thank you for highlighting this, this has been remedied. 

10. The result needs to be more concise, as it is very long.

Response: thank you drawing our attention to this. We have further edited the results section to make it more concise, including moving Table 5 to supplementary data. Please note, that the length of the results section reflects both the large number of included records (51) and the significant heterogeneity of the interventions described. 

In line with methodology of a scoping review the intention is to outline the state of the evidence rather than to identify the most effective form of intervention. Thus, we have endeavoured to succinctly summarise and characterise the widely varying evidence in narrative and table form. We hope that this section is now to your satisfaction, as we believe that this section as it now stands is a significant part of what our review contributes to the understanding of this topic and that if we were to edit it any further it would diminish its usefulness. 

11. Discussion: It needs to be organized according to the most important intervention, as this paper aims to clarify all interventions and identify the most applicable in healthcare.

Recommendations need to be concise, and only applicable recommendations

The paper is very long and needs to be more concise. The authors should choose the most applicable intervention to prevent confusion with the readers.

Response: thank you for this comment. Our paper is a scoping review, and therefore aims to describe the state of the literature rather than to identify the most important or effective intervention. This is reflected in the 4 aims of the scoping review:

“1) To assess and summarise the available interventions for preventing or treating EBS in MHWs

2) To ascertain the theoretical underpinnings and assumptions of these interventions

3) To assess how EBS has been measured in these studies

4) To make recommendations for future research into the treatment and prevention of EBS in MHWs”

Please note, that these aims are consistent with the purpose and function of a scoping review (as opposed to other forms of systematic reviews/meta-analyses). This is described in the Joanna Briggs Scoping Review Framework (Peters et al., 2017). For ease of reference, we have included a relevant excerpt below:

“Unlike systematic reviews, the aim of the scoping reviews is a way of mapping the key concepts that underpin a research area.3 Scoping reviews can be particularly useful for bringing together literature in disciplines with emerging evidence, as they are suited to addressing questions beyond those related to the effectiveness or experience of an intervention. Scoping reviews can be conducted to map a body of literature with relevance to time, location (e.g. country or context), source (e.g. peer-reviewed or grey literature), and origin (e.g. healthcare discipline or academic field).4 The value of scoping reviews to evidence-based practice is the examination of a broader area to identify gaps in the research knowledge base,9 clarify key concepts,10 and report on the types of evidence that address and inform practice in the field… Another distinction between scoping reviews and systematic reviews is that unlike a systematic review, scoping reviews are designed to provide an overview of the existing evidence base regardless of quality.”

With reference to the applicability of included information, please note that as outlined in out methods section, an-priori approach was utilised. In our methods section, we have outlined the steps taken via our search strategy to ensure applicability to MHWs and mental health settings.

12. Consider the previous reviewer comment about the potential conflicts between the needs of health care workers and the needs of the leadership of health care organizations. The authors have stated, "Most studies intervened at the level of the individual, despite the proposed causes of EBS being predominantly organizational." The authors don't address why this is happening. The authors point out that excessive workload is a major contributor to EBS, yet there were no interventions to reduce workload by hiring more healthcare workers. In most circumstances, there are in fact more healthcare workers available to be hired, but doing so is costly. That's the obvious conflict the authors need to address. Organizations usually try to get by with as few workers as possible to save costs. Workers then pay the price by having excessive workloads. If the authors have a different explanation for why organizations don't correct this obvious situation, they should offer it.

Response: thank you for drawing our attention to this point. We agree that attempts at cost saving are no doubt a significant factor in why organisations employing MHWs do not use organisational means to reduce EBS e.g. hiring more staff. We have edited the discussion accordingly. Please see pg40-41- where the below section has been included just after the points about organisational vs individual interventions:

“It is possible that organisational reluctance to prioritise interventions of this nature is driven at least in part by cost-saving concerns, as meaningfully reducing workloads ultimately means hiring more staff. However, evidence suggests that improving staff engagement and satisfaction leads to better care quality, patient satisfaction, financial performance and staff retention (31 ,32), suggesting that, in the long term, this investment is worth the price.”

Review Comments to the Author

Reviewer #1: 

13. I congratulate you for this work you have carried out to fill the gap in the literature. I think that this article will be a precursor for future studies. In addition, the results obtained can contribute to institutional managers taking measures to solve problems, supporting employees and creating awareness of empathy based stress among mental health workers , which is a very important issue.

Response: thank you very much for this feedback, we are delighted by reviewer 1’s comments on the value and important contributions that our paper makes.

---

## [Editor Report · Decision Letter 2]

24 Jun 2024

Interventions to address empathy-based stress in mental health workers: A scoping review and research agenda

PONE-D-23-33814R2

Dear Dr. *Hannah*

We’re pleased to inform you that your manuscript has been judged scientifically suitable for publication and will be formally accepted for publication once it meets all outstanding technical requirements.

Kind regards,

Ebtsam Abou Hashish, 

Academic Editor

PLOS ONE
---

## [Editor Report · Acceptance letter]

25 Oct 2024

PONE-D-23-33814R2 

PLOS ONE

Dear Dr. May, 

I'm pleased to inform you that your manuscript has been deemed suitable for publication in PLOS ONE. Congratulations! Your manuscript is now being handed over to our production team.

Kind regards, 

on behalf of

Prof Ebtsam Aly Omer Abou Hashish 

Academic Editor

PLOS ONE